:ᐱ: PLOS | ONE

**Data Availability Statement:** Data cannot be shared publicly because of privacy of research participants (i.e., data contain potentially identifiable patient information). Restrictions on

# Endothelial dysfunction and low-grade inflammation in the transition to renal replacement therapy

**April C. E. van Gennip**[1☯], **Natascha J. H. Broers**[1,2☯], **Karlien J. ter Meulen**[1], **Bernard Canaud**[3,4], **Maarten H. L. Christiaans**[1], **Tom Cornelis**[5], **Mariëlle A. C. J. Gelens**[1], **Marc M. H. Hermans**[6], **Constantijn J. A. M. Konings**[7], **Jeroen B. van der Net**[1], **Frank M. van der Sande**[1], **Casper G. Schalkwijk**[8,9], **Frank Stifft**[10], **Joris J. J. M. Wirtz**[11], **Jeroen P. Kooman**[1,2]*, **Remy J. H. Martens**[1,9]

1 Department of Internal Medicine, Division of Nephrology, Maastricht University Medical Center+, Maastricht, the Netherlands, 2 NUTRIM School of Nutrition and Translational Research in Metabolism, Maastricht University, Maastricht, the Netherlands, 3 Medical Office EMEA, Fresenius Medical Care Deutschland GmbH, Bad Homburg, Germany, 4 Montpellier University, Montpellier, France, 5 Department of Nephrology, Jessa Hospital, Hasselt, Belgium, 6 Department of Internal Medicine, Division of Nephrology, Viecuri Medical Center, Venlo, the Netherlands, 7 Department of Internal Medicine, Division of Nephrology, Catharina Hospital Eindhoven, Eindhoven, 8 Department of Internal Medicine, Maastricht University Medical Center+, Maastricht, the Netherlands, 9 CARIM School for Cardiovascular Diseases, Maastricht University, Maastricht, the Netherlands, 10 Department of Internal Medicine, Division of Nephrology, Zuyderland Medical Center, Sittard-Geleen, the Netherlands, 11 Department of Internal Medicine, Division of Nephrology, St. Laurentius Hospital, Roermond, the Netherlands

☯ These authors contributed equally to this work.
* jeroen.kooman@mumc.nl

## Abstract

### Introduction

End-stage renal disease (ESRD) strongly associates with cardiovascular disease (CVD) risk. This risk is not completely mitigated by renal replacement therapy. Endothelial dysfunction (ED) and low-grade inflammation (LGI) may contribute to the increased CVD risk. However, data on serum biomarkers of ED and LGI during the transition to renal replacement therapy (dialysis and kidney transplantation) are scarce.

### Methods

We compared serum biomarkers of ED and LGI between 36 controls, 43 participants with chronic kidney disease (CKD) stage 5 non-dialysis (CKD5-ND), 20 participants with CKD stage 5 hemodialysis (CKD5-HD) and 14 participants with CKD stage 5 peritoneal dialysis (CKD5-PD). Further, in 34 and 15 participants repeated measurements were available during the first six months following dialysis initiation and kidney transplantation, respectively. Serum biomarkers of ED (sVCAM-1, E-selectin, P-selectin, thrombomodulin, sICAM-1, sICAM-3) and LGI (hs-CRP, SAA, IL-6, IL-8, TNF-α) were measured with a single- or multiplex array detection system based on electro-chemiluminescence technology.

sharing of such data are imposed by the EU General Data Protection Regulation (GDPR). Data are available for researchers who meet the criteria for access to confidential data, on reasonable request. Data can be requested via Maastricht UMC +, Dept. of Internal Medicine, Div. of Nephrology (secretariaat.nefrologie@mumc.nl) or the principal investigator, Prof. Dr. J. P. Kooman (jeroen. kooman@mumc.nl).

**Funding:** The studies included in this article were supported by an unrestricted grant from Fresenius Medical Care Deutschland GmbH (study NL33129.068.10 and NL43381.068.13), a Baxter International extramural grant (study NL35039.068.10; grant reference number 11CECHHDEU1004) and the Dutch Kidney Foundation (study NL35039.068.10; grant reference number SB166). Bernard Canaud and Adelheid Gauly, who are employees of Fresenius Medical Care, reviewed the manuscript. However, the funding sources had no role in the preparation or analysis of data, and the decision to publish.

**Competing interests:** The studies included in this article were supported by an unrestricted grant from Fresenius Medical Care Deutschland GmbH (study NL33129.068.10 and NL43381.068.13), a Baxter International extramural grant (study NL35039.068.10; grant reference number 11CECHHDEU1004) and the Dutch Kidney Foundation (study NL35039.068.10; grant reference number SB166). BC and AG, who are employees of Fresenius Medical Care and hold shares in the company, reviewed the manuscript. However, the funding sources had no role in the preparation or analysis of data, and the decision to publish. In addition, RJHM, NJHB and JPK are supported by an (un)restricted grant from Fresenius Medical Care. RJHM and NJHB received lectures fees from Fresenius Medical Care. MHC reports Astellas Travel support grants to visit the international conferences ESOT 2017 and TTS 2018. This does not alter our adherence to PLOS ONE policies on sharing data and materials.

# Results

In linear regression analyses adjusted for potential confounders, participants with ESRD had higher levels of most serum biomarkers of ED and LGI than controls. In addition, in CKD5-HD levels of serum biomarkers of ED and LGI were largely similar to those in CKD5-ND. In contrast, in CKD5-PD levels of biomarkers of ED were higher than in CKD5-ND and CKD5-HD. Similarly, in linear mixed model analyses sVCAM-1, thrombomodulin, sICAM-1 and sICAM-3 increased after PD initiation. In contrast, incident HD patients showed an increase in sVCAM-1, P-selectin and TNF-α, but a decline of hs-CRP, SAA and IL-6. Further, following kidney transplantation sVCAM-1, thrombomodulin, sICAM-3 and TNF-α were lower at three months post-transplantation and remained stable in the three months thereafter.

# Conclusions

Levels of serum biomarkers of ED and LGI were higher in ESRD as compared with controls. In addition, PD initiation and, less convincingly, HD initiation may increase levels of selected serum biomarkers of ED and LGI on top of uremia *per se*. In contrast to dialysis, several serum biomarkers of ED and LGI markedly declined following kidney transplantation.

# Introduction

Individuals with end-stage renal disease (ESRD) are at high risk of cardiovascular disease (CVD) mortality[1,2]. Renal replacement therapy is aimed to reduce this risk. However, in the first months after initiation of hemodialysis (HD) an increased mortality risk[3,4] related to CVD[1] is observed. In contrast, kidney transplant recipients have a lower CVD risk as compared with individuals with ESRD who receive dialysis and remain on the waiting list[5,6]. Nevertheless, CVD in kidney transplant recipients risk is still higher than that observed in the general population[6], reflecting a so-called residual CVD risk[7].

Endothelial dysfunction (ED) and low-grade inflammation (LGI) have both been implicated in the adverse CVD risk associated with ESRD[8–10]. Indeed, serum biomarkers of ED and LGI are associated with CVD events in ESRD[11–13], and may be used for CVD risk stratification in individuals with ESRD. In addition, ED and LGI interact at the renal level and may contribute to the development and progression of renal injury, as elegantly described recently [14].

However, to the best of our knowledge, only limited data are available on the course of serum biomarkers of ED and LGI in the transition phase from untreated ESRD to chronic dialysis (*i.e.*, including assessment of ED and LGI prior and following exposure to dialysis)[15,16]. In addition, the available studies were restricted to participants on peritoneal dialysis (PD).

Studies on ED and LGI following kidney transplantation are similarly scarce and either examined LGI only or used a single biomarker to assess ED[17–19].

More extensive assessment of ED and LGI may provide complementary information as both ED and LGI are heterogeneous concepts, which may not be captured with a single biomarker. For example, ED may involve multiple endothelial functions such as regulation of leukocyte adhesion and regulation of hemostasis.

In view of the above, we, first, examined cross-sectional differences in an extensive panel of serum biomarkers of ED and LGI between individuals with chronic kidney disease (CKD)

stage 5 non-dialysis (CKD5-ND; comprising individuals who were scheduled to initiate dialysis or receive a kidney transplant), CKD stage 5 HD (CKD5-HD), CKD stage 5 PD (CKD5-PD) and healthy controls. The serum biomarkers included soluble vascular cellular adhesion molecule 1 (sVCAM-1), E-selectin, P-selectin, thrombomodulin, soluble intercellular adhesion molecule 1 (sICAM-1), soluble intercellular adhesion molecular 3 (sICAM-3), high-sensitivity C-reactive protein (hs-CRP), serum amyloid A (SAA), interleukin 6 (IL-6), interleukin 8 (IL-8), and tumor necrosis factor alpha (TNF-α). Second, we examined the courses of these serum biomarkers during the first six months following initiation of HD and PD, and following kidney transplantation, respectively.

## Materials and methods

### Study population and design

This study consisted of a cross-sectional and a longitudinal part, and included participants from three separate observational studies which have been performed in the southern part of the Netherlands and focused on dialysis initiation (HD and PD), kidney transplantation and chronic dialysis (HD and PD), respectively. These studies were conducted between February 2012 and July 2017, between October 2013 and January 2018, and between June 2012 and December 2017, respectively. The methodology of the individual studies has been published previously [20,21]. A complete overview of in- and exclusion criteria of these studies is also provided in S1 Methods.

S1 Fig is a flowchart showing the derivation of the cross-sectional study population. For the cross-sectional part, we included baseline data of incident dialysis patients and kidney transplant recipients, and data of prevalent dialysis patients. In addition, we included baseline data of healthy kidney donors and data of healthy controls, which together formed a healthy control group. The cross-sectional analyses were performed in the subpopulation with data on serum biomarkers of ED and LGI. Patients receiving nocturnal HD were excluded from the analyses as a previous study that included participants from the chronic dialysis study has already shown that levels of the measured serum biomarkers of ED and LGI are similar to those in conventional HD [21].

Some participants participated in more than one study (*i.e.*, incident dialysis patients who subsequently received a kidney transplant or were studied as prevalent dialysis patients). Only data from their first study was analyzed.

S1 Fig is a flowchart showing the derivation of the longitudinal study population. For the longitudinal part, we included incident HD, incident PD patients and kidney transplant recipients who had complete baseline and follow-up data on serum biomarkers of endothelial dysfunction and low-grade inflammation. Participants who participated in both the study on dialysis initiation and the study on kidney transplantation were included as both dialysis patients and kidney transplant recipients as there was no direct statistical comparison between groups.

Incident HD and PD patients were followed prospectively with assessments at six months after dialysis initiation. Kidney transplant recipients were followed prospectively with assessments at three and six months post-transplantation.

Written informed consent was obtained from each patient prior to participation. The studies were approved by the Ethical Committee (NL43381.068.13, NL33129.068.10, NL35039.068.10) and the Hospital Board of the Maastricht University Medical Center+. All methods were carried out in accordance with relevant guidelines and regulation. In addition, none of the transplant donors were from a vulnerable population and all donors or next of kin provided written informed consent that was freely given.

## Renal replacement therapy modalities

Detailed information on the dialysis treatments applied in both incident and prevalent dialysis patients and the immunosuppressive protocol of kidney transplant recipients are provided in S1 Methods.

## Study measurements

**Serum biomarkers of endothelial dysfunction and low-grade inflammation.** Blood was collected in sterile 5 ml BD vacutainer SST II Advance tubes. After centrifugation, serum was aspirated and stored at -80˚C until analysis. Before analysis, serum samples where thawed and mixed thoroughly.

sVCAM-1, E-selectin, P-selectin, thrombomodulin, sICAM-1, sICAM-3, hs-CRP, SAA, IL-6, IL-8, and TNF-α were measured with a single- or multiplex array detection system based on electro-chemiluminescence technology (Meso Scale Discovery SECTOR Imager 2400, Gaithersburg, Maryland, USA). All measurements were performed in duplicate. Inter-assay and intra-assay variations for all the markers were <9%, except the inter-assay variation for E-selectin (10.1%), sICAM-1 (10.2%) and IL-6 (14.0%).

**Other laboratory parameters.** Dialysis adequacy (Kt/V) was retrospectively collected from the health record (single-pool Kt/V for HD, weekly Kt/V for PD). In addition, serum creatinine and/or serum β2-microglobulin were measured with routine laboratory measurements at the time of study participation, except for healthy kidney donors and kidney transplant recipients, for whom these data were retrospectively collected from the health record. In controls and in participants with CKD5-ND, GFR was estimated with the CKD-EPI equation based on serum creatinine (eGFR$_{\text{CKD-EPI}}$) [22]. In participants with CKD5-D, residual GFR was estimated with an equation based on serum β2-microglobulin (eGFR$_{\text{residual}}$)[23].

**Clinical characteristics.** Origin of renal disease was based on the diagnosis as reported in the patient's health record and categorized as nephrosclerosis, glomerulosclerosis, hypertensive nephropathy, renovascular disease, diabetic nephropathy, polycystic kidney disease, IgA nephropathy, glomerulonephritis, other, and unknown. History of diabetes mellitus and cardiovascular disease, and medication use were based on self-report for healthy controls and the health record for patients. Current smoking was based on self-report. Current dialysis modality, history of kidney transplantation, dialysis vintage and residual urine output were collected from the health record. Body mass index (BMI) was calculated as weight divided by height squared. Fluid overload was measured with the Body Composition Monitor (BCM®) (Fresenius Medical Care, Bad Homburg, Germany). Office blood pressure was measured with an electronic sphygmomanometer (Omron M4-I, Omron, Japan).

All participants were requested to be in a fasting state during the measurements. For practical reasons not all patients were in fasting state as requested, for example, individuals with diabetes. In participants on HD, study measurements and blood sampling were performed prior to one of their dialysis sessions (the midweek session for prevalent HD patients; either the first, second or third session for incident HD patients and for non-preemptively transplanted patients prior to KTx).

## Statistical analyses

Characteristics of the study population for the cross-sectional and longitudinal analyses were presented as means with standard deviations (SDs), medians with 25th and 75th percentiles, and numbers with percentages, as appropriate.

Unadjusted post-hoc comparisons of characteristics of participant groups were performed with the independent Student t-test for normally distributed data, Dunn's test for non-

normally distributed data and Fisher's exact test for categorical data if a one-way analysis of variance (ANOVA) test, Kruskal-Wallis test and Fisher's exact test, respectively, indicated an overall difference between groups.

Correlations among baseline levels of serum biomarkers of ED and LGI were examined with Spearman's rank correlation coefficients, stratified by participant group.

For the cross-sectional analyses, participants were stratified into controls, CKD5-ND (*i.e.*, baseline data of incident dialysis patients and future kidney transplant recipients), CKD5-HD and CKD5-PD (*i.e.*, prevalent dialysis patients including future kidney transplant recipients who were on dialysis at baseline).

Differences in individual serum biomarkers of ED and LGI between CKD5-ND, CKD5-HD, CKD5-PD and controls were evaluated with linear regression analyses to allow adjustment for potential confounders. Levels of serum biomarkers were positively skewed and, therefore, natural log transformed to obtain acceptably normally distributed residuals. The regression coefficients were exponentiated to obtain the ratio of (geometric mean) levels of the serum biomarkers in CKD5-ND, CKD5-HD and CKD5-PD relative to controls and among ESRD groups.

In addition, standardized composite scores (Z-scores) were calculated for ED and LGI and compared across groups with linear regression analyses. The use of composite scores may increase statistical efficiency and reduce the influence of biological and analytical variability [24]. For this purpose, levels of each individual biomarker were natural log transformed and standardized. Subsequently, the Z-scores were calculated as the standardized average of the standardized scores of the individual serum biomarkers. The Z-score for ED included sVCAM-1, E-selectin, P-selectin, thrombomodulin, sICAM-1 and sICAM-3, and the Z-score for LGI consisted of hs-CRP, SAA, IL-6, IL-8, TNF-$\alpha$, sICAM-1 and sICAM-3 [25]. The serum biomarkers sICAM-1 and sICAM-3 were included in the Z-scores for both ED and LGI as both are expressed by endothelial cells and leukocytes [25].

All linear regression analyses were adjusted for age, sex and diabetes mellitus.

Longitudinal analyses were conducted stratified for incident HD and PD patients, and for kidney transplant recipients. Courses of natural log transformed serum biomarkers of ED and LGI over time following dialysis initiation and kidney transplantation were analyzed with linear mixed models to account for within-person correlations between repeated measurements and missing data. The fixed-effects part contained time as categorical variable (baseline served as reference), and the random-effects parts included a random intercept. The models for incident HD and PD patients also contained an interaction term between time and dialysis modality to test whether courses differ between HD and PD. Models were fitted by restricted maximum likelihood. Regression coefficients were exponentiated to obtain the ratio of (geometric mean) biomarker levels relative to baseline levels.

A two-tailed *P* value < 0.050 was considered statistically significant, except for interaction analyses for which 0.100 was used as a cut-off.

Analyses were conducted in IBM SPSS Statistics Version 24.0 (IBM Corp., Armonk, NY, USA) and in R Version 3.5.1 with RStudio Version 1.1.456 combined with the packages tidyverse, haven, scales, ggpubr, corrplot and nlme.

## Results

### Population characteristics—Cross-sectional analyses

Thirty-six controls, 43 participants with CKD5-ND, 20 participants with CKD5-HD and 14 participants with CKD5-PD comprised the cross-sectional study population (Table 1; flowchart in S1 Fig). In general, participants with CKD5-ND, CKD5-HD and CKD5-PD were more often men and had a worse CVD risk profile than controls.

**Table 1. Population characteristics cross-sectional analyses.**

| | Controls | CKD5-ND | CKD5-HD | CKD5-PD |
|---|---|---|---|---|
| | (n = 36) | (n = 43) | (n = 20) | (n = 14) |
| *Clinical characteristics* | | | | |
| Age (years) | 57.6 ±12.3 | 58.8 ±13.9 | 61.4 ±13.6 | 56.9 ±11.8 |
| Men | 19 (52.8%) | 29 (67.4%) | 14 (70.0%) | 9 (64.3%) |
| Origin of end-stage renal disease: | | | ‡ | ‡ |
| Nephrosclerosis | NA | 6 (14.0%) | 0 (0.0%) | 0 (0.0%) |
| Glomerulosclerosis | NA | 2 (4.7%) | 0 (0.0%) | 0 (0.0%) |
| Hypertensive nephropathy | NA | 2 (4.7%) | 4 (20.0%) | 4 (28.6%) |
| Renovascular disease | NA | 0 (0.0%) | 1 (5.0%) | 1 (7.1%) |
| Diabetic nephropathy | NA | 2 (4.7%) | 5 (25.0%) | 2 (14.3%) |
| Polycystic kidney disease | NA | 11 (25.6%) | 6 (30.0%) | 0 (0.0%) |
| IgA nephropathy | NA | 2 (4.7%) | 0 (0.0%) | 1 (7.1%) |
| Glomerulonephritis | NA | 4 (9.3%) | 2 (10.0%) | 4 (28.6%) |
| Nephrotic syndrome | NA | 6 (14.0%) | 0 (0.0%) | 0 (0.0%) |
| Other | NA | 1 (2.3%) | 2 (10.0%) | 1 (7.1%) |
| Unknown | NA | 7 (16.3%) | 0 (0.0%) | 1 (7.1%) |
| History of KTx | NA | 8 (18.6%) | 3 (15.0%) | 3 (21.4%) |
| Immunosuppressive therapy in participants with positive history of KTx | | | | |
| None | NA | 0 (0.0%) | 1 (33.3%) | 1 (33.3%) |
| Prednisolone monotherapy | NA | 1 (12.5%) | 0 (0.0%) | 0 (0.0%) |
| TAC monotherapy | NA | 4 (50.0%) | 1 (33.3%) | 1 (33.3%) |
| MMF monotherapy | NA | 0 (0.0%) | 1 (33.3%) | 1 (33.3%) |
| TAC/MMF monotherapy (unclear) | NA | 1 (12.5%) | 0 (0.0%) | 0 (0.0%) |
| Unknown | NA | 2 (25.0%) | 0 (0.0%) | 0 (0.0%) |
| First future treatment modality | | | | |
| HD | NA | 19 (44.2%) | NA | NA |
| PD | NA | 18 (41.9%) | NA | NA |
| Preemptive KTx | NA | 6 (14.0%) | NA | NA |
| Non-preemptive KTx | NA | 0 (0%) | NA | NA |
| Dialysis vintage (months)* | NA | NA | 27 [22–54] | 14 [6–22] ¶ |
| Kt/V HD (single-pool)/ PD (weekly)* | NA | NA | 2.2 ±0.1 | 2.3 ±0.7 |
| Serum creatinine (μmol/L)* | 83 [75–90] | 579 [431–720] | 722 [565–829] | 694 [495–1106] |
| eGFR$_{CKD-EPI}$ (mL/min/1.73m$^2$)* | 81.4 ±13.6 | 8.6 ±3.1[†] | NA | NA |
| eGFR$_{residual}$ (mL/min/1.73m$^2$)* | NA | NA | 3.6 [1.9–4.4] | 9.3 [4.7–18.8] ¶ |
| Diuresis / Residual urine output* | 16 (100%) | 38 (100%) | 15 (75.0%)‡ | 13 (92.9%) |
| Diuresis / Residual urine output (mL/24h)* | 1,790 [1,138–2,352] | 2,000 [1,700–2,296] | 1,025 [15–1,694] | 1,125 [288–1,533] |
| Diabetes mellitus | 0 (0.0%) | 5 (11.6%) | 8 (40.0%)[†,‡] | 5 (35.7%)[†] |
| Cardiovascular disease | 2 (5.6%) | 13 (32.6%)[†] | 9 (45.0%)[†] | 2 (14.3%) |
| Current smoking | 2 (5.6%) | 8 (18.6%) | 5 (25.0%) | 3 (21.4%) |
| BMI (kg/m$^2$) | 26.0 ±4.1 | 24.8 ±3.9 | 28.1 ±4.6‡ | 27.9 ±4.7‡ |
| Fluid overload (L) | 0.04 ±1.1 | 1.3 ±2.0[†] | 1.2 ±1.5[†] | 1.7 ±2.0[†] |
| SBP (mmHg) | 138.9 ±15.1 | 147.1 ±23.2[†] | 156.6 ±24.6[†] | 156.9 ±28.5[†] |
| DBP (mmHg) | 85.4 ±10.1 | 83.0 ±13.1 | 80.6 ±10.6 | 87.5 ±11.2 |
| Renin-angiotensin-aldosterone system inhibitor use | 2 (5.6%) | 18 (41.9%) | 10 (50.0%) | 11 (78.6%) |

(*Continued*)

**Table 1.** (Continued)

| | Controls | CKD5-ND | CKD5-HD | CKD5-PD |
|---|---|---|---|---|
| | **(n = 36)** | **(n = 43)** | **(n = 20)** | **(n = 14)** |
| Statin use | 2 (5.6%) | 22 (51.2%) | 9 (45.0%) | 10 (71.4%) |

Data are presented as n (%), mean ± standard deviation, or median [25th percentile– 75th percentile].

Abbreviations: AU, arbitrary units; BMI, body mass index; CKD5-D, chronic kidney disease stage 5 dialysis; CKD5-ND, chronic kidney disease stage 5 non-dialysis; DBP, diastolic blood pressure; $eGFR_{CKD-EPI}$, estimated glomerular filtration rate based on the creatinine CKD-EPI equation; $eGFR_{residual}$, estimated residual GFR based on β2-microglobulin; HD, hemodialysis; hs-CRP, high-sensitivity C-reactive protein; IL-6, interleukin 6; IL-8, interleukin 8; KTx, kidney transplantation; MMF, mycophenolate mofetil; NA, not applicable; PD, peritoneal dialysis; SAA, serum amyloid A; SBP, systolic blood pressure; sICAM-1, soluble intercellular adhesion molecule 1; sICAM-3, soluble intercellular adhesion molecule 3; sVCAM-1, soluble vascular cell adhesion molecule 1; TAC, tacrolimus; TNF-α, tumor necrosis factor alpha.

\* Available in (Controls/ CKD5-ND/ CKD5-HD/ CKD-5-PD) n = NA/ NA/ 16/ 11 for dialysis vintage, n = NA/ NA/ 13/ 11 for Kt/V, n = 36/ 42/ 20/ 14 for serum creatinine, n = 36/ 42/ NA/ NA for $eGFR_{CKD-EPI}$, n = NA/ NA/ 15/ 10 for $eGFR_{residual}$, n = 22/ 44/ 20/ 14 for diuresis / residual urine output (dichotomous), n = 14/ 27/ 20/ 14 for diuresis / residual urine output (continuous).

† $P$ value < 0.05 vs. controls;

‡ $P$ value < 0.05 vs. CKD5-ND;

⁋ $P$ value < 0.05 vs. CKD5-HD.

Unadjusted post-hoc comparisons of controls, CKD5-ND, CKD5-HD and CKD5-PD were performed with the independent Student t-test for normally distributed data, Dunn's test for non-normally distributed data and Fisher's exact test for categorical data if an ANOVA, Kruskal-Wallis test and Fisher's exact test, respectively, indicated an overall difference between groups.

Participants with CKD5-HD and CKD5-PD more often had diabetes mellitus than those with CKD5-ND. All participants with CKD5-ND and most with CKD5-HD or CKD5-PD had residual urine output. Participants with CKD5-PD were slightly younger, less often men, less often had a history of CVD, and had higher residual GFR ($eGFR_{residual}$), slightly more fluid overload and shorter dialysis vintage than participants with CKD5-HD, although not all differences were statistically significant in this relatively small study population.

## Correlations among serum biomarkers

In general, the strength of correlations among serum biomarkers of ED and among serum biomarkers of LGI varied. In addition, there was no strict distinction between serum biomarkers of ED and serum biomarkers of LGI in view of the degree of correlation. Further, for some serum biomarkers, correlations in the dialysis groups were in the opposite direction as compared with those in CKD5-ND and controls (S2 Fig).

## Associations of end-stage renal disease with serum biomarkers of endothelial dysfunction and low-grade inflammation

S1 Table and Fig 1 shows the distributions of serum biomarkers of ED and LGI stratified by participant group.

As compared with controls and after adjustment for age, sex and diabetes mellitus (Table 2), sVCAM-1, E-selectin, thrombomodulin, hs-CRP, SAA, IL-6, and TNF-α were higher in CKD5-ND, CKD5-HD and CKD5-PD. In addition, sICAM-1 was higher in CKD5-ND and CKD5-PD, and sICAM-3 was higher in CKD5-PD only.

These results of individual serum biomarkers were also reflected by higher Z-scores for ED and LGI in CKD5-ND, CKD5-HD and CKD5-PD.

As compared with CKD5-ND and after adjustment for age, sex and diabetes mellitus (Table 2), biomarkers of ED and LGI were in general similar in CKD5-HD. In addition, as

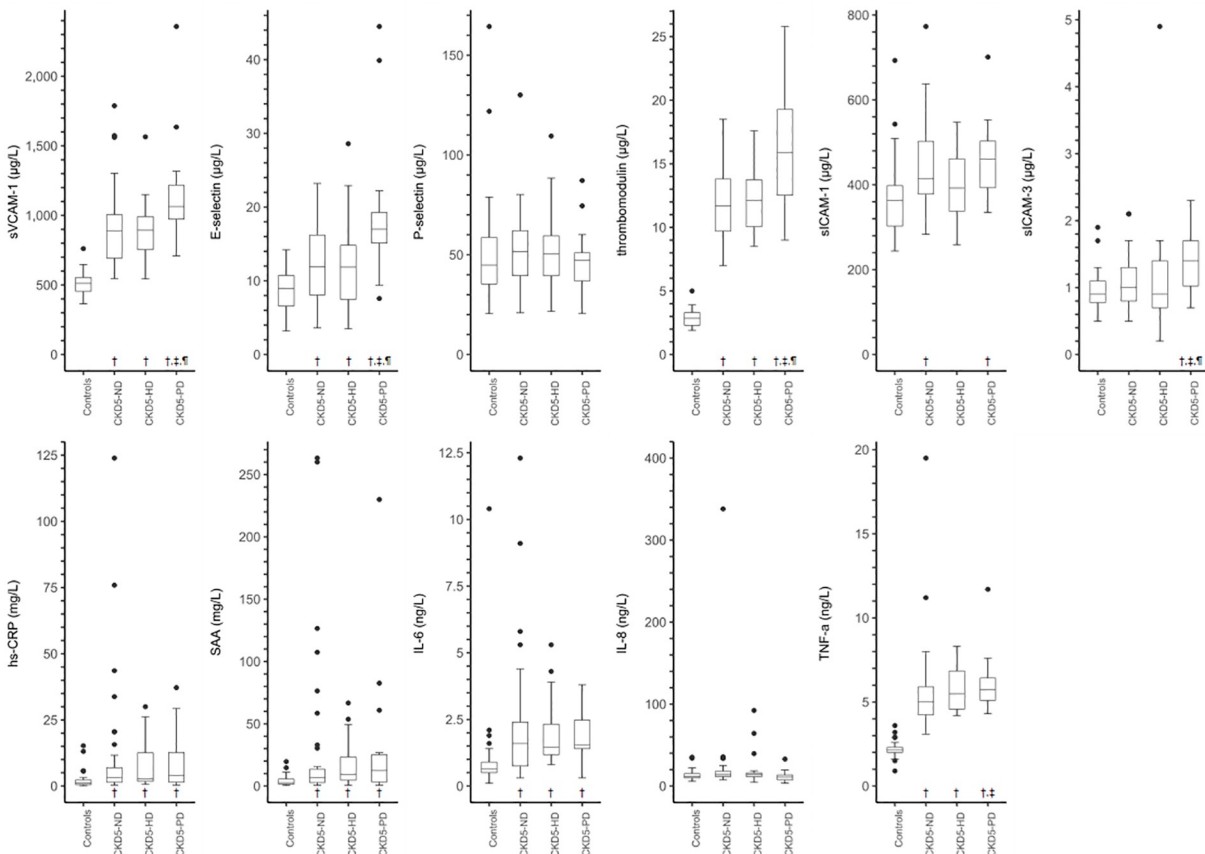

**Fig 1. Boxplots of serum biomarkers of endothelial dysfunction and low-grade inflammation stratified according to participant group.** Abbreviations: hs-CRP, high-sensitivity C-reactive protein; IL-6, interleukin 6; IL-8, interleukin 8; NA, not applicable; SAA, serum amyloid A; sICAM-1, soluble intercellular adhesion molecule 1; sICAM-3, soluble intercellular adhesion molecule 3; sVCAM-1, soluble vascular cell adhesion molecule 1; TNF-α, tumor necrosis factor alpha. Please note that in the boxplot of IL-8 levels in controls, one extreme outlier with an IL-8 level of 2469.8 ng/L was removed to improve the clarity of the graph. † $P$ value < 0.05 vs. controls; ‡ $P$ value < 0.05 vs. CKD5-ND; ¶ $P$ value < 0.05 vs. CKD5-HD based on unadjusted linear regression analyses of natural log transformed serum biomarkers.

compared with both CKD5-ND and CKD5-HD, sVCAM-1, E-selectin, thrombomodulin, sICAM-3 were higher in CKD5-PD. Further, as compared with CKD5-HD, sICAM-1 was higher in CKD5-PD.

These results of individual serum biomarkers were also reflected by a higher Z-score for ED, but not LGI, in CKD5-PD as compared with CKD5-ND and CKD5-HD.

The above results were not materially changed after additional adjustment for systolic or diastolic blood pressure, use of renin-angiotensin-aldosterone system inhibitors, use of HMG-CoA inhibitors, current smoking or BMI (results not shown).

## Population characteristics—Longitudinal analyses

Thirty-four participants with CKD5-ND who initiated dialysis (18 HD, 16 PD), 6 with CKD5-ND who received a kidney transplant, and 5 on chronic dialysis who received a kidney transplant were included in the longitudinal analyses (S2 Table; S1 Fig). In addition, 4 participants who initiated dialysis and later received a kidney transplant were included as kidney transplant recipients as well.

As compared with kidney transplant recipients, incident HD and PD patients were older (65.1 ±12.0 and 57.1 ±12.1 years vs. 51.6 ±12.8 years) and more often had a history of CVD

**Table 2. Associations of end-stage renal disease with serum biomarkers of endothelial dysfunction and low-grade inflammation.**

| Individual serum biomarkers | CKD5-ND vs. controls* | | CKD5-HD vs. controls* | | CKD5-PD vs. controls* | |
|---|---|---|---|---|---|---|
| | Ratio (95%CI) | P value | Ratio (95%CI) | P value | Ratio (95%CI) | P value |
| sVCAM-1 | 1.70 (1.52; 1.90) | < 0.001 | 1.62 (1.40; 1.88) | < 0.001 | 2.08 (1.77; 2.44) | < 0.001 |
| E-selectin | 1.34 (1.10; 1.63) | 0.004 | 1.30 (1.00; 1.68) | 0.048 | 1.98 (1.49; 2.63) | < 0.001 |
| P-selectin | 1.04 (0.87; 1.25) | 0.631 | 1.08 (0.85; 1.37) | 0.534 | 0.93 (0.71; 1.20) | 0.559 |
| Thrombomodulin | 4.04 (3.62; 4.52) | < 0.001 | 4.18 (3.62; 4.83) | < 0.001 | 5.45 (4.65; 6.39) | < 0.001 |
| sICAM-1 | 1.19 (1.08; 1.32) | < 0.001 | 1.08 (0.95; 1.23) | 0.252 | 1.26 (1.09; 1.45) | 0.002 |
| sICAM-3 | 1.14 (0.95; 1.37) | 0.161 | 1.12 (0.88; 1.42) | 0.354 | 1.61 (1.23; 2.09) | < 0.001 |
| hs-CRP | 3.14 (1.73; 5.72) | < 0.001 | 3.28 (1.49; 7.21) | < 0.001 | 3.28 (1.38; 7.80) | < 0.001 |
| SAA | 2.88 (1.61; 5.15) | < 0.001 | 3.65 (1.70; 7.84) | 0.001 | 4.18 (1.80; 9.67) | 0.001 |
| IL-6 | 2.04 (1.48; 2.80) | < 0.001 | 1.88 (1.24; 2.86) | 0.003 | 1.92 (1.22; 3.05) | 0.006 |
| IL-8 | 1.10 (0.78; 1.56) | 0.579 | 1.08 (0.69; 1.71) | 0.728 | 0.78 (0.69; 1.71) | 0.728 |
| TNF-α | 2.41 (2.13; 2.73) | < 0.001 | 2.72 (2.31; 3.20) | < 0.001 | 2.87 (2.40; 3.43) | < 0.001 |
| **Z-scores** | **Beta (95%CI)** | **P value** | **Beta (95%CI)** | **P value** | **Beta (95%CI)** | **P value** |
| Endothelial dysfunction | 1.34 (1.01; 1.67) | < 0.001 | 1.20 (0.77; 1.63) | < 0.001 | 2.00 (1.52; 2.47) | < 0.001 |
| Low-grade inflammation | 1.22 (0.85; 1.58) | < 0.001 | 1.18 (0.70; 1.66) | < 0.001 | 1.48 (0.95; 2.01) | < 0.001 |
| Individual serum biomarkers | CKD5-HD vs. CKD5-ND* | | CKD5-PD vs. CKD5-ND* | | CKD5-PD vs. CKD5-HD* | |
| | Ratio (95%CI) | P value | Ratio (95%CI) | P value | Ratio (95%CI) | P value |
| sVCAM-1 | 0.95 (0.83; 1.09) | 0.491 | 1.22 (1.05; 1.42) | 0.010 | 1.28 (1.08; 1.52) | 0.004 |
| E-selectin | 0.97 (0.76; 1.23) | 0.786 | 1.47 (1.12; 1.93) | 0.005 | 1.52 (1.13; 2.06) | 0.006 |
| P-selectin | 1.03 (0.83; 1.29) | 0.783 | 0.89 (0.69; 1.14) | 0.337 | 0.86 (0.65; 1.13) | 0.276 |
| Thrombomodulin | 1.03 (0.90; 1.18) | 0.633 | 1.35 (1.16; 1.57) | < 0.001 | 1.30 (1.10; 1.54) | 0.002 |
| sICAM-1 | 0.91 (0.80; 1.02) | 0.110 | 1.06 (0.92; 1.21) | 0.430 | 1.17 (1.00; 1.36) | 0.046 |
| sICAM-3 | 0.98 (0.78; 1.23) | 0.880 | 1.41 (1.10; 1.81) | 0.008 | 1.43 (1.08; 1.90) | 0.012 |
| hs-CRP | 1.04 (0.50; 2.18) | 0.909 | 1.04 (0.46; 2.38) | 0.920 | 1.00 (0.40; 2.49) | 0.999 |
| SAA | 1.27 (0.62; 2.59) | 0.510 | 1.45 (0.65; 3.22) | 0.359 | 1.14 (0.47; 2.77) | 0.765 |
| IL-6 | 0.92 (0.62; 1.36) | 0.684 | 0.94 (0.61; 1.46) | 0.792 | 1.02 (0.63; 1.66) | 0.928 |
| IL-8 | 0.98 (0.64; 1.51) | 0.937 | 0.71 (0.44; 1.15) | 0.158 | 0.72 (0.43; 1.23) | 0.225 |
| TNF-α | 1.13 (0.97; 1.31) | 0.124 | 1.19 (1.00; 1.41) | 0.047 | 1.06 (0.87; 1.27) | 0.572 |
| **Z-scores** | **Beta (95%CI)** | **P value** | **Beta (95%CI)** | **P value** | **Beta (95%CI)** | **P value** |
| Endothelial dysfunction | -0.14 (-0.55; 0.26) | 0.485 | 0.65 (0.20; 1.11) | 0.005 | 0.80 (0.30; 1.30) | 0.002 |
| Low-grade inflammation | -0.03 (0.48; 0.42) | 0.880 | 0.27 (-0.24; 0.77) | 0.293 | 0.30 (-0.25; 0.86) | 0.283 |

Ratios represent the ratio of (geometric mean) levels of the serum biomarkers in the respective end-stage renal disease group relative to controls, relative to individuals with chronic kidney disease stage 5 non-dialysis, chronic kidney disease stage 5 hemodialysis, and chronic kidney disease stage 5 peritoneal dialysis, respectively.

Betas represent the differences in Z-scores for endothelial dysfunction and low-grade inflammation (expressed as standard deviations) between the respective end-stage renal disease group and controls, and among the respective end-stage renal disease groups.

All analyses are adjusted for age, sex and diabetes mellitus.

Abbreviations: CKD5-HD, chronic kidney disease stage 5 hemodialysis; CKD5-ND, chronic kidney disease stage 5 non-dialysis; CKD5-PD, chronic kidney disease stage 5 peritoneal dialysis; hs-CRP, high-sensitivity C-reactive protein; IL-6, interleukin 6; IL-8, interleukin 8; SAA, serum amyloid A; sICAM-1, soluble intercellular adhesion molecule 1; sICAM-3, soluble intercellular adhesion molecule 3; sVCAM-1, soluble vascular cell adhesion molecule 1; TNF-α, tumor necrosis factor alpha.

* Analyses based on (Controls/ CKD5-ND/ CKD5-HD/ CKD5-PD) n = 36/43/20/14.

(38.9% and 31.2% vs. 13.3%). In addition, incident HD patients were more often men (83.3% vs. 60.0%). Estimated $GFR_{CKD-EPI}$ was similar and all participants had residual urine output. Incident dialysis patients on PD were younger (57.1 ±12.1 vs. 65.1 ±12.0 years) and less often men (56.2% vs. 83.3%), and had lower BMI (23.5 ±3.0 vs. 26.6 ±3.8 kg/m$^2$), higher diastolic blood pressure (87.3 ±16.0 vs. 77.4 ±9.2 mmHg), and less fluid overload (0.8 ±1.4 vs. 1.9 ±2.4 L),

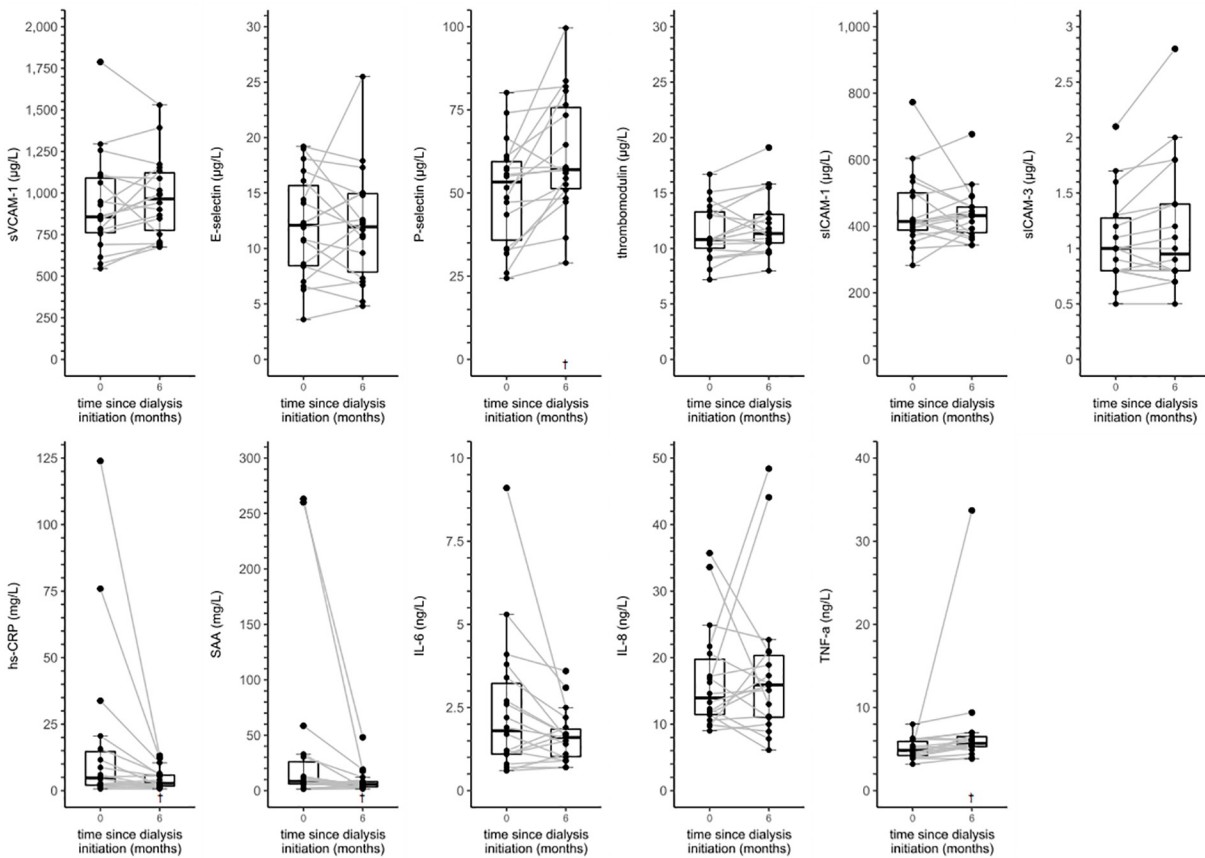

**Fig 2. Boxplots and individual trajectories of serum biomarkers of endothelial dysfunction and low-grade inflammation over time in incident hemodialysis patients.** Abbreviations: hs-CRP, high-sensitivity C-reactive protein; IL-6, interleukin 6; IL-8, interleukin 8; NA, not applicable; SAA, serum amyloid A; sICAM-1, soluble intercellular adhesion molecule 1; sICAM-3, soluble intercellular adhesion molecule 3; sVCAM-1, soluble vascular cell adhesion molecule 1; TNF-α, tumor necrosis factor alpha. [†] $P$ value < 0.05 vs. baseline based on linear mixed model analyses with a random intercept.

although not all differences were statistically significant. One PD patient switched to HD during follow-up.

## Courses of serum biomarkers of endothelial dysfunction and low-grade inflammation following dialysis initiation

Figs 2 and 3 and S3 Table provide data on the distributions and individual trajectories of serum biomarkers of ED and LGI in incident HD and incident PD patients from baseline up to six months after dialysis initiation.

In linear mixed model analyses (Table 3), incident HD patients showed a an increase in sVCAM-1, P-selectin and TNF-α from baseline to six months after dialysis initiation, although this change was only borderline statistically significantly for sVCAM-1. In addition, hs-CRP, SAA and IL-6 were lower after six months of follow-up.

In linear mixed model analyses (Table 3), incident PD patients showed a statistically significant increase in sVCAM-1, thrombomodulin, sICAM-1 and sICAM-3 from baseline to six months after dialysis initiation.

In these analyses, statistically significant interaction terms suggested that the courses of P-selectin, sICAM-1, sICAM-3, hs-CRP, SAA and IL-6 were different between incident HD and incident PD patients ($P_{\text{interaction}}$ < 0.10).

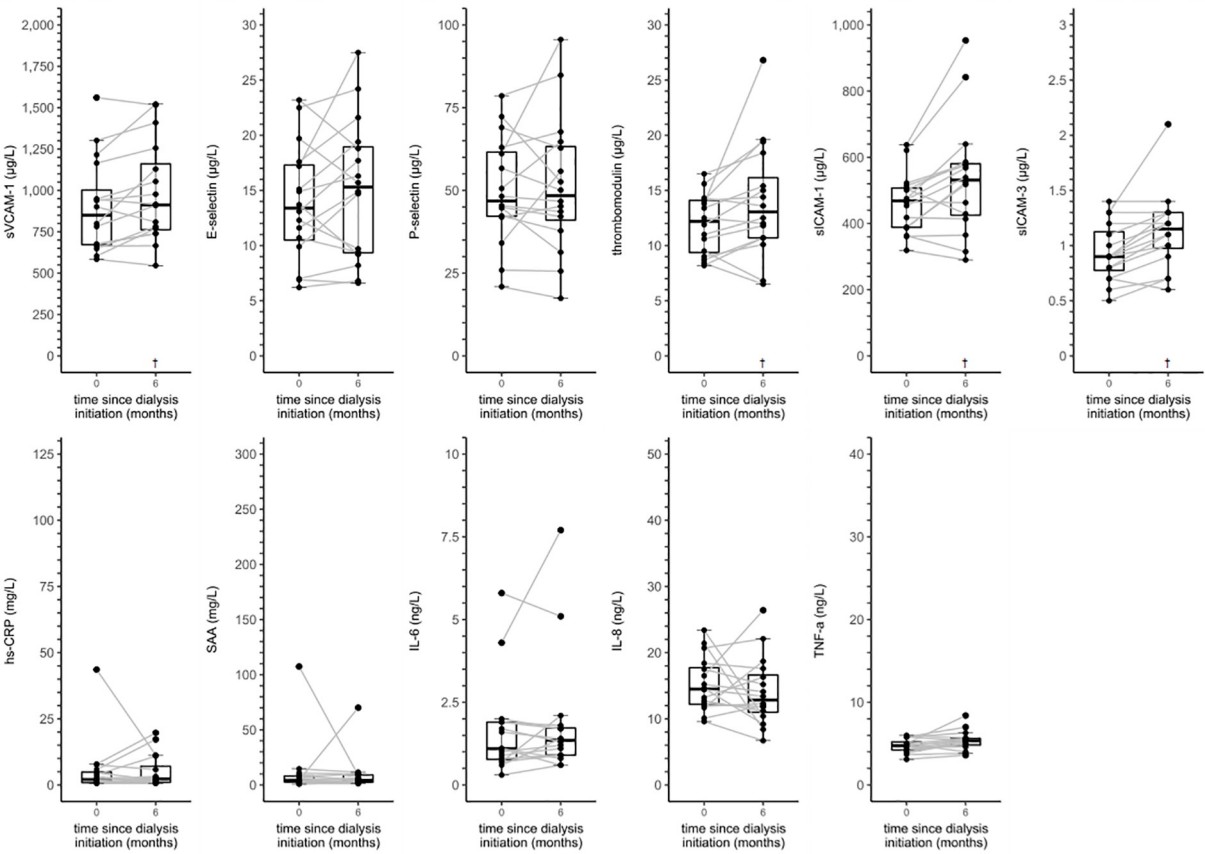

**Fig 3. Boxplots and individual trajectories of serum biomarkers of endothelial dysfunction and low-grade inflammation over time in incident peritoneal dialysis patients.** Abbreviations: hs-CRP, high-sensitivity C-reactive protein; IL-6, interleukin 6; IL-8, interleukin 8; NA, not applicable; SAA, serum amyloid A; sICAM-1, soluble intercellular adhesion molecule 1; sICAM-3, soluble intercellular adhesion molecule 3; sVCAM-1, soluble vascular cell adhesion molecule 1; TNF-α, tumor necrosis factor alpha. [†] $P$ value < 0.05 vs. baseline based on linear mixed model analyses with a random intercept.

### Courses of serum biomarkers of endothelial dysfunction and low-grade inflammation following kidney transplantation

Fig 4 and S4 Table provide data on the distributions and individual trajectories of serum bio-markers of ED and LGI in kidney transplant recipients from baseline up to six months post-transplantation.

In linear mixed model analyses (Table 4), kidney transplant recipients showed a statistically significant reduction in sVCAM-1, thrombomodulin, sICAM-3, and TNF-α at three months post-transplantation, which was still present after six months of follow-up. In addition, E-selectin, hs-CRP, SAA and IL-6 were lower at three and six months post-transplantation with similar or even larger magnitudes of effect (*i.e.*, ratios), albeit not statistically significantly and with wide 95% confidence intervals.

### Additional analyses

First, when we excluded outliers (*i.e.*, defined as participants with standardized residuals < 2 or > 2 SD) in the cross-sectional linear regression analyses, results were largely similar (S5 Table). Most notable dissimilarities after exclusion of outliers were a smaller difference between CKD5-HD and CKD5-PD for E-selectin; more pronounced differences in SAA

**Table 3. Course of serum biomarkers of endothelial dysfunction and low-grade inflammation stratified by dialysis modality.**

| Biomarkers | Modality | Ratios of biomarker following dialysis initiation levels* | | |
| --- | --- | --- | --- | --- |
| | | 6 month vs. baseline | | |
| | | Ratio (95%CI) | P value | $P_{interaction}$** |
| sVCAM-1 (µg/L) | HD | 1.06 (0.99; 1.14) | 0.090 | 0.598 |
| | PD | 1.09 (1.01; 1.18) | 0.023 | |
| E-selectin (µg/L) | HD | 1.00 (0.88; 1.14) | 0.983 | 0.509 |
| | PD | 1.07 (0.93; 1.23) | 0.355 | |
| P-selectin (µg/L) | HD | 1.24 (1.10; 1.38) | < 0.001 | 0.015 |
| | PD | 1.00 (0.89; 1.13) | 0.938 | |
| Thrombomodulin (µg/L) | HD | 1.06 (0.97; 1.15) | 0.224 | 0.335 |
| | PD | 1.12 (1.02; 1.23) | 0.017 | |
| sICAM-1 (µg/L) | HD | 0.98 (0.90; 1.07) | 0.720 | 0.036 |
| | PD | 1.13 (1.03; 1.24) | 0.012 | |
| sICAM-3 (µg/L) | HD | 1.06 (0.98; 1.14) | 0.152 | 0.022 |
| | PD | 1.20 (1.11; 1.30) | < 0.001 | |
| hs-CRP (mg/L) | HD | 0.54 (0.36; 0.81) | 0.004 | 0.015 |
| | PD | 1.14 (0.74; 1.74) | 0.543 | |
| SAA (mg/L) | HD | 0.50 (0.30; 0.82) | 0.008 | 0.057 |
| | PD | 1.01 (0.59; 1.73) | 0.961 | |
| IL-6 (ng/L) | HD | 0.82 (0.65; 1.04) | 0.094 | 0.052 |
| | PD | 1.15 (0.90; 1.46) | 0.261 | |
| IL-8 (ng/L) | HD | 1.02 (0.81; 1.27) | 0.887 | 0.462 |
| | PD | 0.90 (0.71; 1.14) | 0.381 | |
| TNF-α (ng/L) | HD | 1.27 (1.08; 1.49) | < 0.001 | 0.306 |
| | PD | 1.13 (0.95; 1.33) | 0.164 | |

Ratios represent the ratio of (geometric mean) levels of the biomarkers at the respective time point after dialysis initiation relative to baseline levels based on a linear mixed model containing the respective serum biomarkers, categorical time, serum biomarker*categorical time, and a random intercept.

Abbreviations: hs-CRP, high-sensitivity C-reactive protein; IL-6, interleukin 6; IL-8, interleukin 8; NA, not applicable; SAA, serum amyloid A; sICAM-1, soluble intercellular adhesion molecule 1; sICAM-3, soluble intercellular adhesion molecule 3; sVCAM-1, soluble vascular cell adhesion molecule 1; TNF-α, tumor necrosis factor alpha.

* Analyses based on (incident hemodialysis/ incident peritoneal dialysis) n = 18/16.

** P value for the interaction term between categorical time and dialysis modality.

between CKD5-HD/CKD5-PD and CKD5-ND; and more pronounced differences in Z-score for LGI between CKD5-PD and CKD5-ND/CKD5-HD.

Second, when we excluded outliers in analyses for the courses of serum biomarkers following dialysis initiation (S6 Table) and kidney transplantation (S7 Table), results were in general similar or somewhat more pronounced.

Third, results of cross-sectional analyses were similar after exclusion of participants with a history of KTx (results not shown).

Fourth, results of longitudinal analyses were similar for incident HD and PD patients after exclusion of participants with a history of KTx (results not shown). In addition, for kidney transplant recipients, results were largely similar, except for the decline in hs-CRP, SAA and IL-6 which was more pronounced after exclusion of participants with prior KTx (S8 Table).

Fifth, cross-sectional difference in sICAM-3 and the Z-score for ED between CKD5-HD and CKD5-PD were attenuated by ~45% and ~15%, respectively, after additional adjustment for

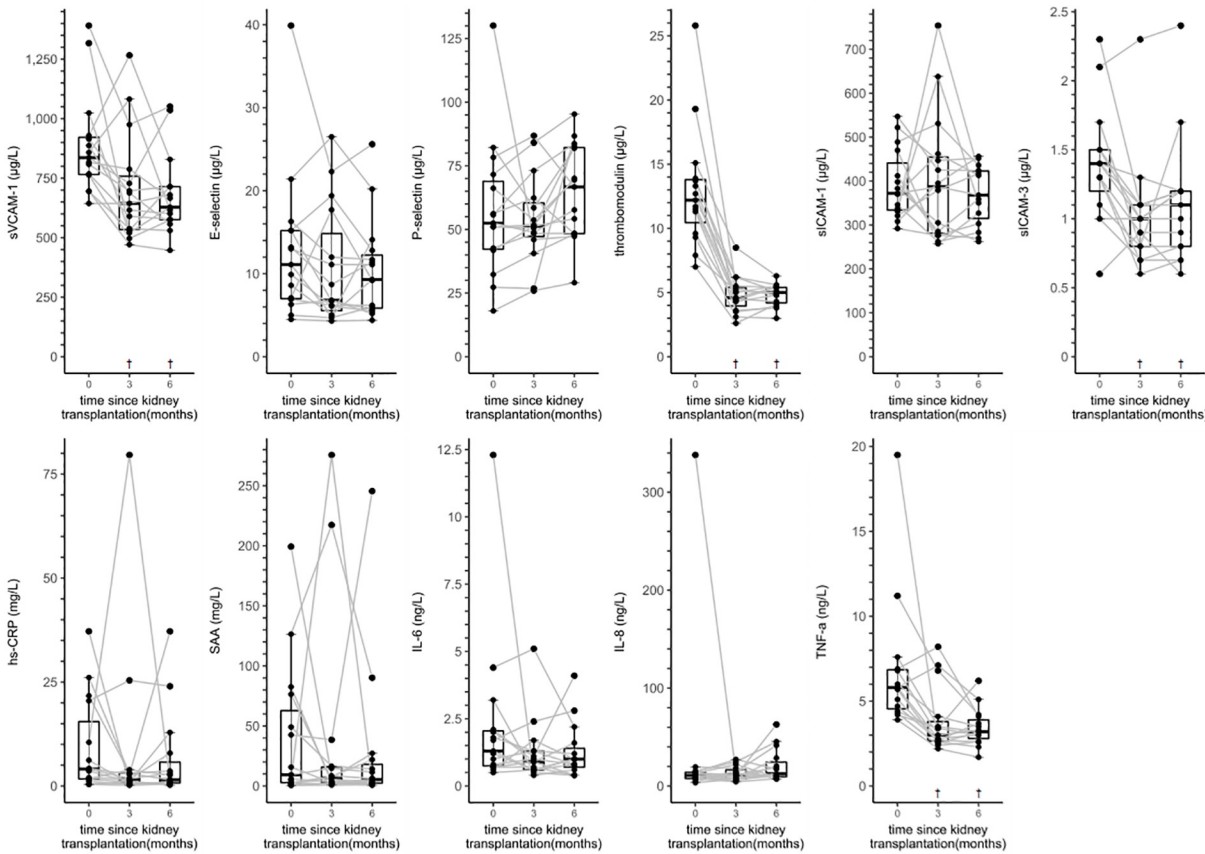

**Fig 4. Boxplots and individual trajectories of serum biomarkers of endothelial dysfunction and low-grade inflammation over time in kidney transplant recipients.** Abbreviations: hs-CRP, high-sensitivity C-reactive protein; IL-6, interleukin 6; IL-8, interleukin 8; NA, not applicable; SAA, serum amyloid A; sICAM-1, soluble intercellular adhesion molecule 1; sICAM-3, soluble intercellular adhesion molecule 3; sVCAM-1, soluble vascular cell adhesion molecule 1; TNF-α, tumor necrosis factor alpha. [†] $P$ value < 0.05 vs. baseline based on linear mixed model analyses with a random intercept.

dialysis vintage. In contrast, results were similar after additional adjustment for eGFR$_{residual}$, residual urine output and fluid overload (results not shown).

Sixth, additional adjustment for eGFR$_{residual}$ in linear mixed model analyses attenuated the courses of sVCAM-1 in incident HD and thrombomodulin in incident PD, but did not explain the courses in the other serum biomarkers (S9 Table).

## Discussion

This study on ED and LGI on individuals transitioning from untreated ESRD to renal replacement therapy had four main findings. First, participants with ESRD had higher levels of most serum biomarkers of ED and LGI than controls. Second, in participants with CKD5-HD, levels of serum biomarkers of ED and LGI were largely similar to those in CKD5-ND. In contrast, participants with CKD5-PD had higher levels of biomarkers of ED than participants with CKD5-ND and CKD5-HD. Third, incident PD patients showed an increase in several serum biomarkers of ED as well, but no convincing change in serum biomarkers of LGI. In contrast, results for incident HD patients were mixed, with an increase in some serum biomarkers of ED and LGI, but a decrease in other biomarkers of LGI. Fourth, following kidney transplantation levels of several serum biomarkers of ED and

**Table 4. Courses of serum biomarkers of endothelial dysfunction and low-grade inflammation following kidney transplantation.**

| Kidney transplant recipients | Ratios of biomarkers following kidney transplantation* | | | |
|---|---|---|---|---|
| | 3 months vs. baseline | | 6 months vs. baseline | |
| Serum biomarkers | Ratio (95%CI) | P value | Ratio (95%CI) | P value |
| sVCAM-1 (µg/L) | 0.78 (0.68; 0.90) | 0.001 | 0.76 (0.66; 0.88) | < 0.001 |
| E-selectin (µg/L) | 0.83 (0.67; 1.02) | 0.075 | 0.85 (0.69; 1.05) | 0.128 |
| P-selectin (µg/L) | 0.99 (0.81; 1.22) | 0.941 | 1.20 (0.98; 1.47) | 0.077 |
| Thrombomodulin (µg/L) | 0.38 (0.32; 0.44) | < 0.001 | 0.39 (0.33; 0.45) | < 0.001 |
| sICAM-1 (µg/L) | 0.99 (0.85; 1.17) | 0.943 | 0.94 (0.80; 1.10) | 0.441 |
| sICAM-3 (µg/L) | 0.73 (0.63; 0.84) | < 0.001 | 0.78 (0.68; 0.91) | < 0.001 |
| hs-CRP (mg/L) | 0.44 (0.18; 1.11) | 0.079 | 0.52 (0.21; 1.29) | 0.152 |
| SAA (mg/L) | 0.76 (0.29; 1.99) | 0.565 | 0.64 (0.24; 1.66) | 0.342 |
| IL-6 (ng/L) | 0.67 (0.40; 1.11) | 0.112 | 0.67 (0.40; 1.11) | 0.112 |
| IL-8 (ng/L) | 0.91 (0.51; 1.61) | 0.732 | 1.30 (0.73; 2.31) | 0.357 |
| TNF-α (ng/L) | 0.57 (0.44; 0.73) | < 0.001 | 0.54 (0.42; 0.69) | < 0.001 |

Ratios represent the ratio of (geometric mean) levels of the biomarkers at the respective time point after kidney transplantation relative to baseline levels based on a linear mixed model containing categorical time and a random intercept.

Abbreviations: hs-CRP, high-sensitivity C-reactive protein; IL-6, interleukin 6; IL-8, interleukin 8; NA, not applicable; SAA, serum amyloid A; sICAM-1, soluble intercellular adhesion molecule 1; sICAM-3, soluble intercellular adhesion molecule 3; sVCAM-1, soluble vascular cell adhesion molecule 1; TNF-α, tumor necrosis factor alpha.

* Analyses based on n = 15.

LGI were lower at three months post-transplantation and remained stable at three months thereafter.

Higher levels of multiple serum biomarkers of ED and LGI in participants with ESRD fits previous literature that characterizes ESRD as a state of ED and LGI[26,27]. In addition, these results indicate that multiple endothelial functions (including regulation of leukocyte adhesion and hemostasis) and inflammatory pathways are affected.

Differences were most remarkable for sVCAM-1, thrombomodulin and TNF-α. In this regard, VCAM-1 is a cell adhesion molecule, which is upregulated by cytokines, including TNF-α[28]. Increased levels of TNF-α may reflect activation of the TNF-α system of peripheral blood monocytes, which could be involved in the immunoincompetence of ESRD patients as well[29]. In addition, increased thrombomodulin levels may indicate ED and/or a hypercoagulable state in ESRD[30].

The causal mechanisms of ED and LGI in ESRD are likely multifactorial and may include exposure to traditional CVD risk factors, oxidative stress, fluid and sodium overload, gut dysbiosis, accumulation of uremic toxins[14,26,31], and turbulent flow[32] due to the hyperdynamic circulation after arteriovenous shunt creation[33]. In addition, ED and LGI may be interrelated[34] as illustrated by the correlation matrix.

Dialysis initiation could either attenuate or augment ED and LGI, which may depend on the relative contributions of improved clearance of uremic toxins and bioincompatibility of the dialysis membrane[35–37].

In this study, both CKD5-PD and incident PD showed an increase in serum biomarkers of ED but not convincingly of serum biomarkers of LGI. In addition, several serum biomarkers of ED were higher in CKD5-PD than in CKD5-HD.

Increases in serum biomarkers of ED following PD initiation have been described previously and hypothesized explanations include alterations in fluid status, mechanical stress of intraperitoneal capillaries and toxicity of absorbed constituents of PD fluid[15]. In this study,

cross-sectional differences between CKD5-HD and CKD-PD were not explained by fluid overload. However, the comparison of HD and PD should be interpreted with caution as the subgroups were small.

For HD it is less clear whether the observed changes reflect effects of dialysis treatment on selected endothelial functions (*i.e.* cellular adhesion) and pathways of LGI, or are the result of the play of chance, as results of the cross-sectional and longitudinal analyses were incongruent. Survivorship bias in prevalent dialysis patients with lower levels of sVCAM-1, P-selectin and TNF-α could be a potential explanation in favor of the results of the longitudinal analyses. In addition, their role as acute phase proteins may confound the courses of hs-CRP, SAA and IL-6.

Kidney transplantation was followed by a reduction in several serum biomarkers of ED and LGI, which was most convincingly shown for sVCAM-1, thrombomodulin, sICAM-3 and TNF-α. The decrease in serum biomarkers of ED agrees with and expands the results of a previous study that assessed ED with sVCAM-1 only[17]. Similarly, the decrease in serum biomarkers of LGI agrees with previous studies on this topic[18,19].

Kidney transplantation may improve ED and LGI by reversing the uremic state that is described above. In this regard, this study indicated that a major part of the benefit of kidney transplantation on ED and LGI is already reached in the first three months post-transplantation. Given their molecular weight, improved renal clearance does not account for the reduction of serum biomarkers of ED but may (partly) explain the reduction of TNF-α[21]. In addition, the role of the immunosuppressive regimen is complex as individual immunosuppressive medications may have opposite effects on ED and LGI. For example, corticosteroids reduce LGI[38] and calcineurin inhibitors induce ED[39], possibly tempering the benefits of kidney transplantation.

In contrast with previous studies, this study observed no statistically significant difference in hs-CRP and IL-6 in the main analyses. This may be explained by a lack of statistical power and high variability as median levels of both serum biomarkers were numerically lower at end of follow-up. In addition, additional analyses suggested that baseline immunosuppressive use in participants with a prior KTx may have confounded results. However, these results should be interpreted with caution given the small sample size. Further, in the present study, levels of hs-CRP and IL-6 were lower as compared with previous studies[18,19]. This may indicate a more favourable prognosis of participants of the present study, which may be related to the higher proportion of pre-emptive kidney transplantations and consequently less dialysis-induced inflammation[40,41].

Strengths of this study were the assessment of multiple serum biomarkers of ED and LGI, and a design that combined a cross-sectional comparison of CKD5-ND, CKD5-HD, CKD5-PD and controls with follow-up data in incident patients on renal replacement therapy. This allowed examining ED and LGI in uremia *per se* as well as the additional short-term consequences of renal replacement therapy. In addition, the use of composite scores on top of an analysis of individual serum biomarkers improved statistical power and reduced biological and analytical variability in the cross-sectional analyses.

This study also had some limitations. First, statistical power was limited, in particular for the comparison of HD and PD, and for the follow-up of kidney transplant recipients. In addition, the limited number of kidney transplant recipients precluded further stratification, for example to compare participants with preemptive and non-preemptive kidney transplantation. Second, the follow-up time was relatively short. However, this study specifically focused on the transition phase from CKD5-ND to CKD5-D and from CKD5-ND to kidney transplant recipient, instead of the long-term course of ED and LGI. In addition, with regard to kidney transplantation, kidney function has stabilized at this point of time, whereas the effects of surgery

are very likely to have been subdued. Third, the distribution of ESRD causes, with varying prognosis, in this study may hamper comparisons between CKD5-ND and CKD5-D patients, and may limit generalizability. For example, the prevalence of polycystic kidney disease was high, whereas that of diabetic nephropathy was low. Fourth, incomparability of characteristics of incident dialysis patients and kidney transplant recipients, which may be related to eligibility for kidney transplantation and, thus, selection bias, hampers a direct comparison between both renal replacement therapies. Fifth, in this observational study, the immunosuppressive regimen was protocolized but not fully standardized, which may hamper the interpretation of results in kidney transplant recipients. Sixth, this study cannot examine whether ED and LGI explain the increased CVD risk in uremia, after dialysis initiation and after kidney transplantation, as data on CVD events were lacking. Nevertheless, the results of this study suggest that it is worthwhile to address this question in a study with data on clinical outcome.

In conclusion, levels of serum biomarkers of ED and LGI were higher in ESRD as compared with controls. In addition, PD initiation and, less convincingly, HD initiation may increase levels of selected serum biomarkers of ED and LGI on top of uremia *per se*. However, the finding that this was not readily apparent for all examined serum biomarkers fits the hypothesis that ED and LGI for a large part already develop in earlier CKD stages. In contrast to dialysis, kidney transplantation led to a marked improvement of several serum biomarkers of ED and LGI.

## Supporting information

**S1 Fig. Flowchart depicting the derivation of the cross-sectional and longitudinal study populations.**
(DOCX)

**S2 Fig. Correlation matrices between serum biomarkers of endothelial dysfunction and low-grade inflammation stratified by participant group.**
(DOCX)

**S1 Methods.**
(DOCX)

**S1 Table. Serum biomarkers of endothelial dysfunction and low-grade inflammation at baseline.**
(DOCX)

**S2 Table. Population characteristics longitudinal analyses.**
(DOCX)

**S3 Table. Courses of serum biomarkers of endothelial dysfunction and low-grade inflammation stratified by dialysis modality.**
(DOCX)

**S4 Table. Courses of serum biomarkers of endothelial dysfunction and low-grade inflammation following kidney transplantation.**
(DOCX)

**S5 Table. Associations of end-stage renal disease with serum biomarkers of endothelial dysfunction and low-grade inflammation after exclusion of individuals with standardized residuals smaller than -2 or larger than 2 standard deviations.**
(DOCX)

**S6 Table. Course of serum biomarkers of endothelial dysfunction and low-grade inflammation stratified by dialysis modality after exclusion of individuals with standardized residuals smaller than -2 or larger than 2 standard deviations.**
(DOCX)

**S7 Table. Courses of serum biomarkers of endothelial dysfunction and low-grade inflammation following kidney transplantation after exclusion of individuals with standardized residuals smaller than -2 or larger than 2 standard deviations.**
(DOCX)

**S8 Table. Courses of serum biomarkers of endothelial dysfunction and low-grade inflammation following kidney transplantation after exclusion of participants with a history of kidney transplantation.**
(DOCX)

**S9 Table. Course of serum biomarkers of endothelial dysfunction and low-grade inflammation stratified by dialysis modality after additional adjustment for estimated residual glomerular filtration rate.**
(DOCX)

## Acknowledgments

We kindly acknowledge Nanda M.P. Diederen for her contribution to the data acquisition.

## Author Contributions

**Conceptualization:** Natascha J. H. Broers, Maarten H. L. Christiaans, Tom Cornelis, Jeroen P. Kooman, Remy J. H. Martens.

**Data curation:** Natascha J. H. Broers, Remy J. H. Martens.

**Formal analysis:** April C. E. van Gennip, Natascha J. H. Broers, Remy J. H. Martens.

**Funding acquisition:** Tom Cornelis, Jeroen P. Kooman.

**Investigation:** Natascha J. H. Broers, Tom Cornelis.

**Methodology:** April C. E. van Gennip, Natascha J. H. Broers, Mariëlle A. C. J. Gelens, Jeroen B. van der Net, Casper G. Schalkwijk, Remy J. H. Martens.

**Project administration:** Natascha J. H. Broers, Maarten H. L. Christiaans, Tom Cornelis.

**Resources:** Maarten H. L. Christiaans, Tom Cornelis, Marc M. H. Hermans, Constantijn J. A. M. Konings, Frank M. van der Sande, Casper G. Schalkwijk, Frank Stifft, Joris J. J. M. Wirtz.

**Supervision:** Natascha J. H. Broers, Mariëlle A. C. J. Gelens, Jeroen B. van der Net, Jeroen P. Kooman, Remy J. H. Martens.

**Visualization:** April C. E. van Gennip, Remy J. H. Martens.

**Writing – original draft:** April C. E. van Gennip, Natascha J. H. Broers, Remy J. H. Martens.

**Writing – review & editing:** Karlien J. ter Meulen, Bernard Canaud, Maarten H. L. Christiaans, Tom Cornelis, Mariëlle A. C. J. Gelens, Marc M. H. Hermans, Constantijn J. A. M. Konings, Jeroen B. van der Net, Frank M. van der Sande, Casper G. Schalkwijk, Frank Stifft, Joris J. J. M. Wirtz, Jeroen P. Kooman.

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
