## [Decision Letter · Decision Letter 0]

12 Jul 2019

PONE-D-19-18004

Endothelial dysfunction and low-grade inflammation in the transition to renal replacement therapy

PLOS ONE

Dear Dr. April van Gennip,

Thank you for submitting your manuscript to PLOS ONE. After careful consideration, we feel that it has merit but does not fully meet PLOS ONE’s publication criteria as it currently stands. Therefore, we invite you to submit a revised version of the manuscript that addresses the points raised during the review process.

ACADEMIC EDITOR: Although it is of interest, the reviewers have raised a number of points which we believe major modifications are necessary to improve the manuscript, taking into account the reviewers' remarks. 

We would appreciate receiving your revised manuscript by Aug 26 2019 11:59PM. To enhance the reproducibility of your results, we recommend that if applicable you deposit your laboratory protocols in protocols.io, where a protocol can be assigned its own identifier (DOI) such that it can be cited independently in the future. For instructions see: http://journals.plos.org/plosone/s/submission-guidelines#loc-laboratory-protocols

We look forward to receiving your revised manuscript.

Kind regards,

Wisit Cheungpasitporn, MD, FACP, FASN

University of Mississippi Medical Center

Twitter: @wisit661 Email: wcheungpasitporn@gmail.com 

Academic Editor

PLOS ONE

Journal Requirements:

2. In your Methods section, please provide additional information regarding the source of donated organs received by the patients in the study. Please specify whether the study involved the use of donated tissue/organs from any vulnerable populations, and provide information on the consent given by the donor or their next of kin. Examples of vulnerable populations include prisoners, subjects with reduced mental capacity due to illness or age, and children. If such a population was used, please ensure you have describe the population and justify the decision to use tissue/organ donations from this group. If not, please state in your Ethics Statement, 'None of the transplant donors were from a vulnerable population and all donors or next of kin provided written informed consent that was freely given.

The studies included in this article were supported by an unrestricted grant from Fresenius Medical Care Deutschland GmbH (study NL33129.068.10 and NL43381.068.13), a Baxter International extramural grant (study NL35039.068.10; grant reference number 11CECHHDEU1004) and the Dutch Kidney Foundation (study NL35039.068.10; grant reference number SB166). Bernard Canaud and Adelheid Gauly, who are employees of Fresenius Medical Care, reviewed the manuscript. However, the funding sources had no role in the preparation or analysis of data, and the decision to publish.

We note that you received funding from a commercial source: Fresenius Medical Care Deutschland GmbH

Reviewers' comments:

Reviewer's Responses to Questions

**Comments to the Author**

1. Is the manuscript technically sound, and do the data support the conclusions?

Reviewer #1: No

Reviewer #2: Partly

Reviewer #3: Yes

2. Has the statistical analysis been performed appropriately and rigorously? 

Reviewer #1: I Don't Know

Reviewer #2: I Don't Know

Reviewer #3: Yes

3. Have the authors made all data underlying the findings in their manuscript fully available?

Reviewer #1: No

Reviewer #2: Yes

Reviewer #3: Yes

4. Is the manuscript presented in an intelligible fashion and written in standard English?

Reviewer #1: No

Reviewer #2: Yes

Reviewer #3: Yes

5. Review Comments to the Author

Reviewer #1: The manuscript is of interest providing some interesting data on the evolution of biomarkers of endothelial function and inflammation, but many concerns are present.

General comments: The manuscript is very difficult to read with a tendency to be confused. Statistical analysis must be reviewed by a specialist.

The abstract must be reviewed. The methods and results are not clear and the conclusion I is not really conclusive. Word (completely between the brackets is not a good idea in a scientific paper.

Ref 3 is of no interest and must be removed.

In methods part, the timing of blood samples collection must be more precise. First the vintage of collection must be given for all the patients. Second, the timing of collection regarding the dialysis séance must be precise, first, second or third dialysis of the week. Third, for the biomarkers, how the sample were conserved? All the measurements were done at the same time? Did the authors have data on the effects of blood conservation on the concentration of the biomarkers?

For others biological value the technique of dosage must be provide.

Finally, the kidney transplantation protocol is very different with three patient included in a randomized trial. I think these patients must be excluded. The values, especially for inflammation markers are very difficult to interpret when one a small sample of patients (15) you have 3 with monotherapy with tacrolimus, 5 with tacro and MMF and 7 with Tacro MMF and corticoids. It is not clear for me for patients with everolimus if they are still with everolimus or not.

Pour les dilyser penser à discuter les différentes membranes.

In results,

In table 1, the authors must separate HD and PD patients. In controls and CKD5-ND it is not residual urine outpout but diuresis. In this table, in the 18% of patient with CKD have a history of Ktx, did they have an immunosuppressive drugs? This must be stated and clear in the characteristics of the patients. It is very different to study endothelial dysfunction to have or not IS drugs.

Important clinical data are missing: tobacco, alcool, lipids concentration, calcium phosphore bicarbonate potassium, hb etc. Medications used, mainly immunosuppressive drugs and corticoids, antihypertensive drugs, HMG CoA inhibitors, anticoagulants, aspirin etc. All this drugs could play an important role on endothelial function and inflammation.

For the table 2, supplementary analysis separating HD and PD must be done. Adjustment with blood pressure could be of interest mainly regarding endothelial dysfunction.

For the longitudinal analysis we must have separation of PD and HD patients. In addition, in my opinion for dialysis patient regarding the boxplot there is no real differences in all the biomarkers, the test used for statistical significance must be provide explicitly in the results. More interesting than a box blot the authors must provide individual courses of the biomarkers separating HD and PD.

We must have comparisons with other biological values, like calcium phosphore, bicarbonate urea, creatininemia etc.

For transplantation patients same remarks regarding the representation of data. It could be of interest to indicate the course of values for the 4 patients undergoing dialysis then transplantation the data.

The additional analysis section is a non sense. All this analysis must be included with main analysis or not discusses. The patients with cyst infection must be definitively excluded.

The discussion will be analyzed after the modification asked for the methods and results.

Reviewer #2: In the present study, the authors compared biomarkers of endothelium dysfunction and low-grade inflammation in controls and in different groups of CKD patients. They show that most biomarkers are higher in CKD patients than in controls, higher in dialysis patients than in non-dialyzed patients and decline after kidney transplantation. Higher levels of serum biomarkers of ED and LGI in CKD patients have been previously shown in the literature. In addition, the data on serum biomarkers after kidney transplantation do not provide new informations and only confirm previous studies.

1) The design of the study is complex. It mixes the data of three separate observational studies, but the number of subjects per group is quite low. The characteristics of CKD5-ND and CKD5-D patients are different in terms of origin of end stage renal disease, and presence of diabetes mellitus.

2) Patients on PD are different from patients on HD (younger, less CVD history, higher residual GFR…), and additional analysis demonstrated differences in biomarkers in these two groups of patients. Therefore,the authors should not include these patients in the same CKD5D group.

3) The choice of statistical analyses is not clear. Why do the authors compare the data of biomarkers between groups with linear regression analysis and not with tests for group comparison like ANOVA or Mann Whitney? In addition, why do the authors use a natural log transformation to perform statistical test on the levels of serum biomarkers? How is the distribution of the data (normal or not)?

4) Identical data are presented several times, in different figures/tables. For example, Figure 2 and Table 3 provides the same informations, data in Table 3 are also in S1 Table etc…

5) For a better understanding, the authors should add the symbols of p values on figures and tables. In particular, the p values between CKD5-ND and CKD5D should be mentioned in Table 1.

6) In table 3, an entire column is NA, please remove the column. In S1 Table, dialysis vintage and KT/V are NA, please remove the lines.

Reviewer #3: Manuscript is well written. Only minor suggestions to improve the manuscript.

Please try to avoid the use of () in the descriptive sentences in academic writing “This risk is not (completely) mitigated by renal replacement therapy”

“Only data of their first study was analyzed” “of” should be “from”

Also, I suggest the investigators to may improve the introduction/discussion by going over very nice review on this topic PMID: 30607032 Nat Rev Nephrol. 2019 Feb;15(2):87-108. doi: 10.1038/s41581-018-0098-z

6. PLOS authors have the option to publish the peer review history of their article (what does this mean?). If published, this will include your full peer review and any attached files.

Reviewer #1: No

Reviewer #2: No

Reviewer #3: No

---

## [Author Response · Author response to Decision Letter 0]

8 Aug 2019

van Gennip et al., PLOS ONE

Manuscript ID: PONE-D-19-18004

August 8, 2019

Dear editor, dear reviewers,

Thank you for providing us an opportunity to revise our manuscript. We thank the editor and the reviewers for their thoughtful comments and suggestions which have helped us to improve our manuscript. For your convenience, please find below, in bold, our point-by-point rebuttal. All changes made in the manuscript are highlighted with ‘tracked changes’. The page numbers of the manuscript are used below to indicate where the changes have been made. 

On behalf of all authors,

Yours sincerely,

April van Gennip, MD Remy Martens, MD, PhD

 

Response to the Reviewer #1

The manuscript is of interest providing some interesting data on the evolution of biomarkers of endothelial function and inflammation, but many concerns are present.

General comments: The manuscript is very difficult to read with a tendency to be confused. Statistical analysis must be reviewed by a specialist. 

We thank the reviewer for the time spend to review the manuscript.

1) The abstract must be reviewed. The methods and results are not clear and the conclusion I is not really conclusive. 

We have reviewed and updated the abstracts within the constraints of the word limits (page 2-3, lines 37-67):

“Introduction: End-stage renal disease (ESRD) strongly associates with cardiovascular disease (CVD) risk. This risk is not completely mitigated by renal replacement therapy. Endothelial dysfunction (ED) and low-grade inflammation (LGI) may contribute to the increased CVD risk. However, data on serum biomarkers of ED and LGI during the transition to renal replacement therapy (dialysis and kidney transplantation) are scarce. 

Methods: We compared serum biomarkers of ED and LGI between 36 controls, 43 participants with chronic kidney disease (CKD) stage 5 non-dialysis (CKD5-ND), 20 participants with CKD stage 5 hemodialysis (CKD5-HD) and 14 participants with CKD stage 5 peritoneal dialysis (CKD5-PD). Further, in 34 and 15 participants repeated measurements were available during the first six months following dialysis initiation and kidney transplantation, respectively. Serum biomarkers of ED (sVCAM-1, E-selectin, P-selectin, thrombomodulin, sICAM-1, sICAM-3) and LGI (hs-CRP, SAA, IL-6, IL-8, TNF-α) were measured with a single- or multiplex array detection system based on electro-chemiluminescence technology.

Results: In linear regression analyses adjusted for potential confounders, participants with ESRD had higher levels of most serum biomarkers of ED and LGI than controls. In addition, in CKD5-HD levels of serum biomarkers of ED and LGI were largely similar to those in CKD5-ND. In contrast, in CKD5-PD levels of biomarkers of ED were higher than in CKD5-ND and CKD5-HD. Similarly, in linear mixed model analyses sVCAM-1, thrombomodulin, sICAM-1 and sICAM-3 increased after PD initiation. In contrast, incident HD patients showed an increase in sVCAM-1, P-selectin and TNF-α, but a decline of hs-CRP, SAA and IL-6. Further, following kidney transplantation sVCAM-1, thrombomodulin, sICAM-3 and TNF-α were lower at three months post-transplantation and remained stable in the three months thereafter.

Conclusions: Levels of serum biomarkers of ED and LGI were higher in ESRD as compared with controls. In addition, PD initiation and, less convincingly, HD initiation may increase levels of selected serum biomarkers of ED and LGI on top of uremia per se. In contrast to dialysis, several serum biomarkers of ED and LGI markedly declined following kidney transplantation.”

2) Word (completely between the brackets is not a good idea in a scientific paper.

We have removed the brackets as requested. Abstract (page 2, lines 39): “This risk is not (completely) mitigated by renal replacement therapy.”

3) Ref 3 is of no interest and must be removed.

We agree. This reference was inserted erroneously and we have now removed this reference.

4) In methods part, the timing of blood samples collection must be more precise. First the vintage of collection must be given for all the patients. 

It is not entirely clear what the reviewer means with ‘vintage of collection’. All samples were collected during the period the study was conducted. This is described in the methods section in the paragraph study population and design. The studies included were conducted between February 2012 and July 2017, between October 2013 and January 2018, and between June 2012 and December 2017, respectively. 

5) Second, the timing of collection regarding the dialysis séance must be precise, first, second or third dialysis of the week.

In the group of prevalent dialysis patients samples were collected prior to each patients’ mid-week dialysis session. In the group of incident dialysis patients the timing of blood sampling relative to the dialysis session was not standardized (i.e., either the first, second or third dialysis of the week). The same held true for kidney transplant patients who were on dialysis prior to KTx. 

We have added this information to the Methods section (page 9, lines 194-198): “In participants on HD, study measurements and blood sampling were performed prior to one of their dialysis sessions (the midweek session for prevalent HD patients; either the first, second or third session for incident HD patients and for non-preemptively transplanted patients prior to KTx).”

6) Third, for the biomarkers, how the sample were conserved? All the measurements were done at the same time? Did the authors have data on the effects of blood conservation on the concentration of the biomarkers?

Blood was collected in sterile 5 ml BD vacutainer SST II Advance tubes. After centrifugation, serum was aspirated and stored at -80°C until analysis. Before analysis, serum samples where thawed and mixed thoroughly. 

We have added this information to the Methods (page 8, lines 156-158):” Blood was collected in sterile 5 ml BD vacutainer SST II Advance tubes. After centrifugation, serum was aspirated and stored at -80°C until analysis. Before analysis, serum samples where thawed and mixed thoroughly.”

All measurements were done at the same time. Biomarkers were analyzed with 6 multiplex assays. Per assay, analyses were performed within the same batch to avoid batch effects. In addition, combining the analyses of two assays reduced the number of freeze-thaw cycles to three in total. 

Unfortunately, data on long-term stability of the serum biomarkers are lacking, except for hs-CRP, which is stable up to 11 years if stored at -80°C [1].

7) For others biological value the technique of dosage must be provide. 

It is not clear to us what the reviewer means with ‘technique of dosage’. 

8) Finally, the kidney transplantation protocol is very different with three patient included in a randomized trial. I think these patients must be excluded.

All patients received standard protocol patient care during the kidney transplant period. The only difference is the use of different immunosuppressive protocols. However, the immunosuppressive medication as described in the manuscript and used in the RCT (TRANSFORM) and the observational study are already part of standard patient care. Therefore, we chose to include these patients into the analyses since our study has an observational nature and as such therapy cannot be fully standardized. 

9) The values, especially for inflammation markers are very difficult to interpret when one a small sample of patients (15) you have 3 with monotherapy with tacrolimus, 5 with tacro and MMF and 7 with Tacro MMF and corticoids. It is not clear for me for patients with everolimus if they are still with everolimus or not.

The immunosuppressive regimen, albeit protocolized, is tailored towards the individual patient’s clinical response, leading to some variation in immunosuppressive medication. This is inherent to clinical practice and the observational nature of this study.

However, we have mentioned this as a study limitation in the Discussion (page 32, lines 544-546): “Fifth, in this observational study, the immunosuppressive regimen was protocolized but not fully standardized, which may hamper the interpretation of results in kidney transplant recipients.

Two patients started with everolimus. One patient was lossed-to-follow-up, the other patient used everolimus throughout the entire study period. Which is now stated more clearly in the S1 methods: “Three patients included in this study were simultaneously enrolled in a different study (TRANSFORM) (NCT01950819). Two of these three patients were randomized to receive tacrolimus, everolimus and prednisolone as triple therapy from time of kidney transplant. One of these two patients withdrew informed consent for further follow-up in our study in the first week after kidney transplantation for personal reasons. The other received everolimus throughout the entire study period. The third patient was randomized to receive tacrolimus (prograft), MMF and prednisolone as triple therapy from time of kidney transplant. This patient switched to the local immunosuppressive protocol of our centre after the 6 month follow up visit.”

10) Pour les dilyser penser à discuter les différentes membranes. 

It is not entirely clear what the reviewer means with this. An overview of the dialysis membranes used for the patients in this study is provided in S1 Methods (paragraph on dialysis therapy modalities).

11) In results, In table 1, the authors must separate HD and PD patients. In controls and CKD5-ND it is not residual urine outpout but diuresis. 

Thank you for noticing. We have changed ‘residual urine output’ into ‘diuresis/ residual urine output’ in Table 1 and S1 Table. In addition, we now present characteristics in Table 1 stratified according to controls, CKD5-ND, CKD5-HD and CKD5-PD. Similarly, we now present characteristics in S2 Table stratified according to incident HD, incident PD and kidney transplant recipients.

The description of the cross-sectional population characteristics in the Results (page 13, lines 257-266) has been adapted as well: “Thirty-six controls, 43 participants with CKD5-ND, 20 participants with CKD5-HD and 14 participants with CKD5-PD comprised the cross-sectional study population (Table 1; flowchart in S1 Fig). In general, participants with CKD5-ND, CKD5-HD and CKD5-PD were more often men and had a worse CVD risk profile than controls.

Participants with CKD5-HD and CKD5-D more often had diabetes mellitus than those with CKD5-ND. All participants with CKD5-ND and most with CKD5-D had residual urine output. Participants with CKD5-PD slightly younger, less often men, less often had a history of CVD, and had higher residual GFR (eGFRresidual), slightly more fluid overload and shorter dialysis vintage than participants with CKD5-HD, although not all differences were statistically significant in this relatively small study population.”

Similarly for the description of the longitudinal population characteristics (page 20, lines 334-348): “As compared with kidney transplant recipients, incident HD and PD patients were older (65.1 ±12.0 and 57.1 ±12.1 years vs. 51.6 ±12.8 years) and more often had a history of CVD (38.9% and 31.2% vs. 13.3%). In addition, incident HD patients were more often men (83.3% vs. 60.0%). Estimated GFRCKD-EPI was similar and all participants had residual urine output. Incident dialysis patients on PD were younger (57.1 ±12.1 vs. 65.1 ±12.0 years) and less often men (56.2% vs. 83.3%), and had lower BMI (23.5 ±3.0 vs. 26.6 ±3.8 kg/m2), higher diastolic blood pressure (87.3 ±16.0 vs. 77.4 ±9.2 mmHg), and less fluid overload (0.8 ±1.4 vs. 1.9 ±2.4 L), although not all differences were statistically significant. One PD patient switched to HD during follow-up.”

12) In this table, in the 18% of patient with CKD have a history of Ktx, did they have an immunosuppressive drugs? This must be stated and clear in the characteristics of the patients. It is very different to study endothelial dysfunction to have or not IS drugs.

We agree that this information is relevant. Therefore, we have added data on immunosuppressive medication use in participants with a history of KTx in both cross-sectional and longitudinal analyses, where available, in Table and S2 Table. In addition, we have added more detailed information to S1 Methods.

S1 Methods:

“For the 14 participants who were included in the cross-sectional analyses and had a history of kidney transplantation, data on their immunosuppressive regimen were retrospectively collected from the electronic patient health record. Of the 8 participants with CKD5-ND and a history of kidney transplantation, 7 participants were on immunosuppressive therapy of whom the therapeutic regimen could be retrieved for 6 participants: 1 prednisolone monotherapy, 4 TAC monotherapy, 1 either TAC or MMF monotherapy (not entirely clear from the health record). Of the 3 participants with CKD5-HD and a history of kidney transplantation, 1 was on TAC monotherapy, 1 on MMF monotherapy and 1 did not use immunosuppressive therapy. Similarly, of the 3 participants with CKD5-PD and a history of kidney transplantation, 1 was one TAC monotherapy, 1 on MMF monotherapy and 1 did not use immunosuppressive therapy.

All incident HD patients with a history of kidney transplantation were on immunosuppressive medication: 2 TAC monotherapy from baseline throughout follow-up, 1 either TAC or MMF monotherapy (not entirely clear from the health record) from baseline up to and including the six month follow-up measurement, and for 1 of these participants the exact immunosuppressive regimen could not be retrieved.

Similarly, all incident PD patients with a history of kidney transplantation were on immunosuppressive medication: 1 prednisolone from baseline throughout follow-up and 1 TAC monotherapy from baseline up to and including the six months follow-up measurement.

All kidney transplant recipients with a history of kidney transplantation were on immunosuppressive therapy: 1 TAC monotherapy and 2 MMF monotherapy.”

Further, we have performed sensitivity analyses excluding participants with prior KTx to explore confounding effects of immunosuppressive use and the presence of a KTx transplant. For cross-sectional analyses, results were similar. In addition, for longitudinal analyses, results were similar for incident HD and PD patients. Further, for kidney transplant recipients, results were largely similar, except for the decline in hs-CRP, SAA and IL-6, which was more pronounced.

We have reported these results as follows in the additional analyses section of the Results (page 27, lines 432-433):

“Third, results of cross-sectional analyses were similar after exclusion of participants with a history of KTx (results not shown).

Fourth, results of longitudinal analyses were similar for incident HD and PD patients after exclusion of participants with a history of KTx (results not shown). In addition, for kidney transplant recipients, results were largely similar, except for the decline in hs-CRP, SAA and IL-6 which was more pronounced after exclusion of participants with prior KTx (S8 Table).” 

13) Important clinical data are missing: tobacco, alcool, lipids concentration, calcium phosphore bicarbonate potassium, hb etc. Medications used, mainly immunosuppressive drugs and corticoids, antihypertensive drugs, HMG CoA inhibitors, anticoagulants, aspirin etc. All this drugs could play an important role on endothelial function and inflammation.

We agree with the reviewer that addition of these data would improve the manuscript. Therefore, we have added smoking status, use of renin-angiotensin-aldosterone system inhibitors and HMG CoA inhibitors to Table 1 and S1 Table. The use of immunosuppressive medication is already described in detail for kidney transplant recipients in S1 Methods. Unfortunately, data on the suggested laboratory values, which may provide explanatory information for the associations reported, as well as data on anticoagulant and aspirin use have not been collected for the present study. 

Methods (page 9, lines 184-185): “Current smoking was based on self-report.”

In addition, in line with the comment, we explored whether additional adjustment for systolic blood pressure, diastolic blood pressure, use of renin-angiotensin-aldosterone system inhibitors, use of HMG CoA inhibitors, current smoking and bmi would change results: results were similar.

Results (page 16, lines 307-309): “The above results were not materially changed after additional adjustment for systolic or diastolic blood pressure, use of renin-angiotensin-aldosterone system inhibitors, use of HMG-CoA inhibitors, current smoking or BMI (results not shown). “

14) For the table 2, supplementary analysis separating HD and PD must be done. Adjustment with blood pressure could be of interest mainly regarding endothelial dysfunction.

We now present the analyses in Table 2 stratified for CKD5-HD and CKD5-PD.

Accordingly, the Results section (page 16, lines 292-306) has been adapted as well:

“Table S1 and Figure 1 shows the distributions of serum biomarkers of ED and LGI stratified by participant group.

As compared with controls and after adjustment for age, sex and diabetes mellitus (Table 2), sVCAM-1, E-selectin, thrombomodulin, hs-CRP, SAA, IL-6, and TNF-α were higher in CKD5-ND, CKD5-HD and CKD5-PD. In addition, sICAM-1 was higher in CKD5-ND and CKD5-PD, and sICAM-3 was higher in CKD5-PD only.

These results of individual serum biomarkers were also reflected by higher Z-scores for ED and LGI in CKD5-ND, CKD5-HD and CKD5-PD.

As compared with CKD5-ND and after adjustment for age, sex and diabetes mellitus (Table 2), biomarkers of ED and LGI were in general similar in CKD5-HD. In addition, as compared with both CKD5-ND and CKD5-HD, sVCAM-1, E-selectin, thrombomodulin, sICAM-3 were higher in CKD5-PD. Further, as compared with CKD5-HD, sICAM-1 was higher in CKD5-PD. 

These results of individual serum biomarkers were also reflected by a higher Z-score for ED, but not LGI, in CKD5-PD as compared with CKD5-ND and CKD5-HD.”

In addition, we have adjusted the first paragraph of the Discussion (page 29, lines 499-453):

“First, participants ESRD had higher levels of most serum biomarkers of ED and LGI than controls. Second, in participants with CKD5-HD, levels of serum biomarkers of ED and LGI were largely similar to those in CKD5-ND. In contrast, participants with CKD5-PD had higher levels of biomarkers of ED than participants with CKD5-ND and CKD5-HD.”

Additional analyses with adjustment for systolic or diastolic blood pressure as well as renin-angiotensin-aldosterone system inhibitor use, HMG CoA use, current smoking and BMI, other import risk factors for ED, were similar. We have added this the Results section (page 16, lines 207-309): “The above results were not materially changed after additional adjustment for systolic or diastolic blood pressure, use of renin-angiotensin-aldosterone system inhibitors, use of HMG-CoA inhibitors, current smoking or BMI (results not shown).”

15) For the longitudinal analysis we must have separation of PD and HD patients. In addition, in my opinion for dialysis patient regarding the boxplot there is no real differences in all the biomarkers, the test used for statistical significance must be provide explicitly in the results. More interesting than a box blot the authors must provide individual courses of the biomarkers separating HD and PD.

Longitudinal analyses stratified by dialysis modality were in the supplementary analyses. We have now moved these analyses to the main manuscript (Table 2). In addition, we have added individual trajectories to the box plots as requested, both for incident HD patients (Fig 2) and incident PD patients (Fig 3).

Accordingly, the Results section (page 20-21, lines 352-365.) has been adapted as well: 

“Figure 2, Figure 3 and S3 Table show the distributions of serum biomarkers of ED and LGI in incident HD and incident PD patients from baseline up to six months after dialysis initiation.

In linear mixed model analyses (Table 3), incident HD patients showed an increase in sVCAM-1, P-selectin and TNF-α from baseline to six months after dialysis initiation, although this change was only borderline statistically significantly for sVCAM-1. In addition, hs-CRP, SAA and IL-6 were lower after six months of follow-up.

In linear mixed model analyses (Table 3), incident PD patients showed a statistically significant increase in sVCAM-1, thrombomodulin, sICAM-1 and sICAM-3 from baseline to six months after dialysis initiation. 

In these analyses, statistically significant interaction terms suggested that the courses of P-selectin, sICAM-1, sICAM-3, hs-CRP, SAA and IL-6 were different between incident HD and incident PD patients (Pinteraction < 0.10).”

Please note that we have reanalyzed the data with the more modern linear mixed model technique. In contrast to repeated measures ANOVA, linear mixed model analysis provides an effect size (here, ratios of concentrations relative to baseline and corresponding 95% confidence interval) in addition to a test of statistical significance, which is presented in the Tables as well. Further, it allows selecting outliers based in regression diagnostics to assess the robustness of results (see below).

This is addressed in the Methods section (page 11, lines 239-249):

“Longitudinal analyses were conducted stratified for incident HD and PD patients, and for kidney transplant recipients. Courses of natural log transformed serum biomarkers of ED and LGI over time following dialysis initiation and kidney transplantation were analyzed with linear mixed models to account for within-person correlations between repeated measurements and missing data. The fixed-effects part contained time as categorical variable (baseline served as reference), and the random-effects parts included a random intercept. The models for incident HD and PD patients also contained an interaction term between time and dialysis modality to test whether courses differ between HD and PD. Models were fitted by restricted maximum likelihood. Regression coefficients were exponentiated to obtain the ratio of (geometric mean) biomarker levels relative to baseline levels.”

In addition, we have adjusted the first paragraph of the Discussion (page 29, lines 453-457):

“Third, incident PD patients showed an increase in several serum biomarkers of ED as well, but no convincing change in serum biomarkers of LGI. In contrast, results for incident HD patients were mixed, with an increase in some serum biomarkers of ED and LGI, but a decrease in other biomarkers of LGI.”

16) We must have comparisons with other biological values, like calcium phosphore, bicarbonate urea, creatininemia etc.

We agree that these data would improve the manuscript. Unfortunately, however, these data were not available for the present study, except for serum creatinine, which we have added to Table 1 and S2 Table (please see also our response on comment 13).

17) For transplantation patients same remarks regarding the representation of data. It could be of interest to indicate the course of values for the 4 patients undergoing dialysis then transplantation the data.

Similar to the presentation of data of dialysis patients, we now provide boxplots with individual trajectories for kidney transplant recipients (Fig 4). In addition, we have reanalyzed the data with linear mixed model analyses as well.

This has been adapted in the Results section (page 24, lines 299-404):

“In linear mixed model analyses (Table 4), kidney transplant recipients showed a statistically significant reduction in sVCAM-1, thrombomodulin, sICAM-3, and TNF-α at three months post-transplantation, which was still present after six months of follow-up. In addition, E-selectin, hs-CRP, SAA and IL-6 were lower at three and six months post-transplantation with similar or even larger magnitudes of effect (i.e., ratios), albeit not statistically significantly and with wide 95% confidence intervals.

In addition, we have adjusted the first paragraph of the Discussion (page 29, lines 457-459):

“Fourth, following kidney transplantation levels of several serum biomarkers of ED and LGI were lower at three months post-transplantation and remained stable at three months thereafter.”

Although we agree that a comparison of preemptively and non-preemptively transplanted patients would be interesting, the statistical power of the present study is too low to perform such an analyses and risk of chance findings is high. Instead, we have mentioned this as study limitation (page 32, lines 528-532): “First, statistical power was limited, in particular for the comparison of HD and PD, and for the follow-up of kidney transplant recipients. In addition, the limited number of kidney transplant recipients precluded further stratification, for example to compare participants with preemptive and non-preemptive kidney transplantation.”

18) The additional analysis section is a non sense. All this analysis must be included with main analysis or not discusses. The patients with cyst infection must be definitively excluded.

The correlation plots (page 13, lines 268-274) and stratified analyses for HD vs. PD have now been included in the main analyses (page 13-23, lines 256-392). In addition, the explorative analyses on associations of eGFRCKD-EPI, eGFRresidual, residual urine output and dialysis vintage have been removed, as requested.

We respectfully disagree with the reviewer that the patient with a history of recurring cyst infection should be excluded a priori. Indeed, his most recent episode was six months before study participation and a recent transluminal angioplasty of the patient’s dialysis shunt may be an alternative explanation for the increased inflammatory parameters [2]. Nevertheless, we agree that this was not clear from our formulation in the previous version of the manuscript. In addition, other participants may have had a rise in inflammatory markers due to an intercurrent infection or inflammatory response as well. Therefore, we have now more performed sensitivity analyses after exclusion of outliers (defined as standardized residuals < -2 or > 2 SD on the respective analysis). In general, results were largely similar after exclusion of outliers.

Results (page 27, lines 422-431): 

“First, when we excluded outliers (i.e., defined as participants with standardized residuals < 2 or > 2 SD) in the cross-sectional linear regression analyses, results were largely similar (S5 Table). Most notable dissimilarities after exclusion of outliers were a smaller difference between CKD5-HD and CKD5-PD for E-selectin; more pronounced differences in SAA between CKD5-HD/CKD5-PD and CKD5-ND; and more pronounced differences in Z-score for LGI between CKD5-PD and CKD5-ND/CKD5-HD.

Second, when we excluded outliers in analyses for the courses of serum biomarkers following dialysis initiation (S6 Table) and kidney transplantation (S7 Table), results were in general similar or somewhat more pronounced.”

In addition, we consider the additional adjustments for eGFRresidual, residual urine output, dialysis vintage meaningful: these analyses explore potential explanations for differences between CKD5-HD and CKD5-ND, but may distract from the main message if placed in the main results.

Therefore, we have retained these results as part of the additional analyses (page 27, lines 439-443).

In addition, the use of linear mixed model analyses allowed exploring whether the course of eGFRresidual explained the courses of the serum biomarkers following HD or PD initiation. As we think this question naturally follows from the results, we have added it to the additional analyses (page 28, lines 444-446): “Sixth, additional adjustment for eGFRresidual in linear mixed model analyses attenuated the courses of sVCAM-1 in incident HD and thrombomodulin in incident PD, but did not explain the courses in the other serum biomarkers (S9 Table).”

19) The discussion will be analyzed after the modification asked for the methods and results.

We have incorporated the results of the stratified analyses of HD and PD throughout the Discussion.

 

Response to the Reviewer #2 

In the present study, the authors compared biomarkers of endothelium dysfunction and low-grade inflammation in controls and in different groups of CKD patients. They show that most biomarkers are higher in CKD patients than in controls, higher in dialysis patients than in non-dialyzed patients and decline after kidney transplantation. Higher levels of serum biomarkers of ED and LGI in CKD patients have been previously shown in the literature. In addition, the data on serum biomarkers after kidney transplantation do not provide new informations and only confirm previous studies.

We thank the reviewer for the time spend to review the manuscript.

1) The design of the study is complex. It mixes the data of three separate observational studies, but the number of subjects per group is quite low. The characteristics of CKD5-ND and CKD5-D patients are different in terms of origin of end stage renal disease, and presence of diabetes mellitus.

We agree with the reviewer that the design is complex. To help the reader in appreciating the flow of patients of the respective study groups, we have provided a flow diagram in S1 Fig. 

The limited study power is addressed as study limitation (page 32, lines 528-532): “. First, statistical power was limited, in particular for the comparison of HD and PD, and for the follow-up of kidney transplant recipients. In addition, the limited number of kidney transplant recipients precluded further stratification, for example to compare participants with preemptive and non-preemptive kidney transplantation.”

We agree that characteristics of CKD5-ND and CKD5-D patients differ in terms to cause of ESRD and prevalence of diabetes mellitus. Therefore, comparisons between CKD5-ND and CKD5-D were adjusted for diabetes mellitus. Unfortunately, the number of participants was too low to stratify by cause of ESRD. We added this to the study limitations (page 32, lines 537-540): “Third, the distribution of ESRD causes, with varying prognosis, in this study may hamper comparisons between CKD5-ND and CKD5-D patients, and may limit generalizability. For example, the prevalence of polycystic kidney disease was high, whereas that of diabetic nephropathy was low.”

2) Patients on PD are different from patients on HD (younger, less CVD history, higher residual GFR…), and additional analysis demonstrated differences in biomarkers in these two groups of patients. Therefore,the authors should not include these patients in the same CKD5D group.

We have now provided results stratified by dialysis modality, both for the cross-sectional and longitudinal analyses. The Results section, including Tables and Figures, has been adapted accordingly.

Results (page 13, lines 256-266):

“Population characteristics – cross-sectional analyses

Thirty-six controls, 43 participants with CKD5-ND, 20 participants with CKD5-HD and 14 participants with CKD5-PD comprised the cross-sectional study population (Table 1; flowchart in S1 Fig). In general, participants with CKD5-ND, CKD5-HD and CKD5-PD were more often men and had a worse CVD risk profile than controls.

Participants with CKD5-HD and CKD5-D more often had diabetes mellitus than those with CKD5-ND. All participants with CKD5-ND and most with CKD5-D had residual urine output. Participants with CKD5-PD slightly younger, less often men, less often had a history of CVD, and had higher residual GFR (eGFRresidual), slightly more fluid overload and shorter dialysis vintage than participants with CKD5-HD, although not all differences were statistically significant in this relatively small study population.”

 

Results (page 16, lines 292-306):

“Associations of end-stage renal disease with serum biomarkers of endothelial dysfunction and low-grade inflammation

Table S1 and Figure 1 shows the distributions of serum biomarkers of ED and LGI stratified by participant group.

As compared with controls and after adjustment for age, sex and diabetes mellitus (Table 2), sVCAM-1, E-selectin, thrombomodulin, hs-CRP, SAA, IL-6, and TNF-α were higher in CKD5-ND, CKD5-HD and CKD5-PD. In addition, sICAM-1 was higher in CKD5-ND and CKD5-PD, and sICAM-3 was higher in CKD5-PD only.

These results of individual serum biomarkers were also reflected by higher Z-scores for ED and LGI in CKD5-ND, CKD5-HD and CKD5-PD.

As compared with CKD5-ND and after adjustment for age, sex and diabetes mellitus (Table 2), biomarkers of ED and LGI were in general similar in CKD5-HD. In addition, as compared with both CKD5-ND and CKD5-HD, sVCAM-1, E-selectin, thrombomodulin, sICAM-3 were higher in CKD5-PD. Further, as compared with CKD5-HD, sICAM-1 was higher in CKD5-PD. 

These results of individual serum biomarkers were also reflected by a higher Z-score for ED, but not LGI, in CKD5-PD as compared with CKD5-ND and CKD5-HD.”

Results (page 20, lines 333-348):

“Population characteristics – longitudinal analyses

Thirty-four participants with CKD5-ND who initiated dialysis (18 HD, 16 PD), 6 with CKD5-ND who received a kidney transplant, and 5 on chronic dialysis who received a kidney transplant were included in the longitudinal analyses (S2 Table; S1 Fig). In addition, 4 participants who initiated dialysis and later received a kidney transplant were included as kidney transplant recipients as well. 

As compared with kidney transplant recipients, incident HD and PD patients were older (65.1 ±12.0 and 57.1 ±12.1 years vs. 51.6 ±12.8 years) and more often had a history of CVD (38.9% and 31.2% vs. 13.3%). In addition, incident HD patients were more often men (83.3% vs. 60.0%). Estimated GFRCKD-EPI was similar and all participants had residual urine output. Incident dialysis patients on PD were younger (57.1 ±12.1 vs. 65.1 ±12.0 years) and less often men (56.2% vs. 83.3%), and had lower BMI (23.5 ±3.0 vs. 26.6 ±3.8 kg/m2), higher diastolic blood pressure (87.3 ±16.0 vs. 77.4 ±9.2 mmHg), and less fluid overload (0.8 ±1.4 vs. 1.9 ±2.4 L), although not all differences were statistically significant. One PD patient switched to HD during follow-up.”

Results (page 20-21, lines 350-365):

“Courses of serum biomarkers of endothelial dysfunction and low-grade inflammation following dialysis initiation

Fig 2, Fig 3 and S3 Table provide data on the distributions and individual trajectories of serum biomarkers of ED and LGI in incident HD and incident PD patients from baseline up to six months after dialysis initiation. 

In linear mixed model analyses (Table 3), incident HD patients showed an increase in sVCAM-1, P-selectin and TNF-α from baseline to six months after dialysis initiation, although this change was only borderline statistically significantly for sVCAM-1. In addition, hs-CRP, SAA and IL-6 were lower after six months of follow-up.

In linear mixed model analyses (Table 3), incident PD patients showed a statistically significant increase in sVCAM-1, thrombomodulin, sICAM-1 and sICAM-3 from baseline to six months after dialysis initiation. 

In these analyses, statistically significant interaction terms suggested that the courses of P-selectin, sICAM-1, sICAM-3, hs-CRP, SAA and IL-6 were different between incident HD and incident PD patients (Pinteraction < 0.10).”

In addition, we have adjusted the first paragraph of the Discussion (page 29, lines 448-459):

“This study on ED and LGI on individuals transitioning from untreated ESRD to renal replacement therapy had four main findings. First, participants ESRD had higher levels of most serum biomarkers of ED and LGI than controls. Second, in participants with CKD5-HD, levels of serum biomarkers of ED and LGI were largely similar to those in CKD5-ND. In contrast, participants with CKD5-PD had higher levels of biomarkers of ED than participants with CKD5-ND and CKD5-HD. Third, incident PD patients showed an increase in several serum biomarkers of ED as well, but no convincing change in serum biomarkers of LGI. In contrast, results for incident HD patients were mixed, with an increase in some serum biomarkers of ED and LGI, but a decrease in other biomarkers of LGI. Fourth, following kidney transplantation levels of several serum biomarkers of ED and LGI were lower at three months post-transplantation and remained stable at three months thereafter.”

Further, we have incorporated the results of analyses stratified by dialysis modality throughout the Discussion.

3) The choice of statistical analyses is not clear. Why do the authors compare the data of biomarkers between groups with linear regression analysis and not with tests for group comparison like ANOVA or Mann Whitney? In addition, why do the authors use a natural log transformation to perform statistical test on the levels of serum biomarkers? How is the distribution of the data (normal or not)?

We chose to analyze between-group differences with linear regression analyses as this technique allows adjustment for potential confounders (here a limited selection of age, sex and diabetes mellitus given the low statistical power).

The distribution of serum biomarkers were positively skewed. Therefore, these values were natural log transformed to obtained acceptably normally distributed residuals of the regression analyses.

We have now clarified this in the Methods (page 10, lines 218-224): “Differences in individual serum biomarkers of ED and LGI between CKD5-ND, CKD5-HD, CKD5-PD and controls were evaluated with linear regression analyses to allow adjustment for potential confounders. Levels of serum biomarkers were positively skewed and, therefore, natural log transformed to obtain acceptably normally distributed residuals. The regression coefficients were exponentiated to obtain the ratio of (geometric mean) levels of the serum biomarkers in CKD5-ND, CKD5-HD and CKD5-PD relative to controls and among ESRD groups.”

Please note that we have reanalyzed the data with the more modern linear mixed model technique, which in contrast to repeated measures ANOVA, provides an effect size (here, ratios of concentrations relative to baseline and corresponding 95% confidence interval) in addition to a test of statistical significance, which is presented in the Tables as well.

This is addressed in the Methods section (page 11, lines 239-249):

“Longitudinal analyses were conducted stratified for incident HD and PD patients, and for kidney transplant recipients. Courses of natural log transformed serum biomarkers of ED and LGI over time following dialysis initiation and kidney transplantation were analyzed with linear mixed models to account for within-person correlations between repeated measurements and missing data. The fixed-effects part contained time as categorical variable (baseline served as reference), and the random-effects parts included a random intercept. The models for incident HD and PD patients also contained an interaction term between time and dialysis modality to test whether courses differ between HD and PD. Models were fitted by restricted maximum likelihood. Regression coefficients were exponentiated to obtain the ratio of (geometric mean) biomarker levels relative to baseline levels.”

4) Identical data are presented several times, in different figures/tables. For example, Figure 2 and Table 3 provides the same informations, data in Table 3 are also in S1 Table etc…

We have adapted the tables to reduce overlapping information in the main manuscript. However, we think that some readers prefer tabulated data on serum biomarkers over boxplots. Therefore, we have provided such tables as supplementary data (S1 Table, S3 Table and S4 Table).

5) For a better understanding, the authors should add the symbols of p values on figures and tables. In particular, the p values between CKD5-ND and CKD5D should be mentioned in Table 1.

We have added symbols for P values for comparisons among patient groups to Table 1 as requested and added this to the Methods section as well.

Methods (page 10, lines 206-210):

“Unadjusted post-hoc comparisons of characteristics of participant groups were performed with the independent Student t-test for normally distributed data, Dunn’s test for non-normally distributed data and Fisher’s exact test for categorical data if a one-way analysis of variance (ANOVA) test, Kruskal-Wallis test and Fisher’s exact test, respectively, indicated an overall difference between groups.”

Legend Table 1 (page 15, lines 286-288): “† P value < 0.05 vs. controls; ‡ P value < 0.05 vs. CKD5-ND; ¶ P value < 0.05 vs. CKD5-HD. Unadjusted post-hoc comparisons of controls, CKD5-ND, CKD5-HD and CKD5-PD were performed with the independent Student t-test for normally distributed data, Dunn’s test for non-normally distributed data and Fisher’s exact test for categorical data if an ANOVA, Kruskal-Wallis test and Fisher’s exact test, respectively, indicated an overall difference between groups (†, P value < 0.05 vs. controls; ‡ P value < 0.05 vs. CKD5-ND; ¶ P value < 0.05 vs. CKD5-HD).”

Legend S2 Table: “¶ P value < 0.05 vs. CKD5-HD based on Student’s t test for normally distributed data, Wilcoxon rank sum test for non-normally distributed data and Fisher’s exact test for categorical data.”

In addition, we have added figures for P values to Fig 1-4, as requested.

6) In table 3, an entire column is NA, please remove the column. In S1 Table, dialysis vintage and KT/V are NA, please remove the lines.

We have adapted the Tables as requested. Please note that former Table 3 is now S3 Table and S4 Table, and that former S1 Table is now S2 Table. 

Response to Reviewer #3

Manuscript is well written. Only minor suggestions to improve the manuscript.

We thank the reviewer for the compliment and the time spend to review the manuscript.

1) Please try to avoid the use of () in the descriptive sentences in academic writing “This risk is not (completely) mitigated by renal replacement therapy”

We have removed the brackets as requested. 

Abstract (page 2, lines 39-40): “This risk is not (completely) mitigated by renal replacement therapy.”

Methods (page 9, lines 182-184): “History of diabetes mellitus and cardiovascular disease, and medication use were based on self-report for healthy controls and the health record for patients.”

2) “Only data of their first study was analyzed” “of” should be “from”

We have changed this accordingly (page 6, lines 129): “Only data from their first study was analyzed.”

3) Also, I suggest the investigators to may improve the introduction/discussion by going over very nice review on this topic PMID: 30607032 Nat Rev Nephrol. 2019 Feb;15(2):87-108. doi: 10.1038/s41581-018-0098-z

We thank the reviewer for the recommended review. As the review is mostly on the effects of ED at the renal level instead of systemic ED, we briefly mention this in the Introduction (page 4, lines 80-82): “In addition, ED and LGI interact at the renal level and may contribute to the development and progression of renal injury, as elegantly described recently[3].”

In addition, we added this review as a reference to our discussion on the causal mechanisms of ED and LGI in ESRD (page 29-30, lines 470-474): “The causal mechanisms of ED and LGI in ESRD are likely multifactorial and may include exposure to traditional CVD risk factors, oxidative stress, fluid and sodium overload, gut dysbiosis, accumulation of uremic toxins[3-5], and turbulent flow[6] due to the hyperdynamic circulation after arteriovenous shunt creation[7]. In addition, ED and LGI may be interrelated[8] as illustrated by the correlation matrix.”

 

Journal Requirements:

We have adapted the manuscript according to PLOS ONE’s style requirements. The heading of the section listing supporting information was changed to Supporting information (page 38, lines 691). 

2. In your Methods section, please provide additional information regarding the source of donated organs received by the patients in the study. Please specify whether the study involved the use of donated tissue/organs from any vulnerable populations, and provide information on the consent given by the donor or their next of kin. Examples of vulnerable populations include prisoners, subjects with reduced mental capacity due to illness or age, and children. If such a population was used, please ensure you have describe the population and justify the decision to use tissue/organ donations from this group. If not, please state in your Ethics Statement, 'None of the transplant donors were from a vulnerable population and all donors or next of kin provided written informed consent that was freely given.

We have added this to the Methods section (page 7, lines 144-146): In addition, none of the transplant donors were from a vulnerable population and all donors or next of kin provided written informed consent that was freely given.

Donors are carefully provided information and the screening procedure of a potential donor is essential in living kidney donation. Therefore, potential donors are screened and evaluated according to the national Guideline ‘Evaluation of potential donors for living kidney donation’ of the Dutch Transplant Society and the Dutch Kidney Transplantation Advisory Committee. These guidelines are based on the ‘The United Kingdom Guidelines for Living Donor Kidney Transplantation’ of the British Transplantation Society and the Renal Association (see link) 

https://bts.org.uk/wp-content/uploads/2018/01/BTS_LDKT_UK_Guidelines_2018.pdf. 

All donors have been informed about the risks and consequences of kidney donation, and before donation written informed consent is needed. Independent psychosocial evaluation is mandatory to be sure that the potential donor is well informed and understands the risks of donation and a further life with a reduced eGFR.

The studies included in this article were supported by an unrestricted grant from Fresenius Medical Care Deutschland GmbH (study NL33129.068.10 and NL43381.068.13), a Baxter International extramural grant (study NL35039.068.10; grant reference number 11CECHHDEU1004) and the Dutch Kidney Foundation (study NL35039.068.10; grant reference number SB166). Bernard Canaud and Adelheid Gauly, who are employees of Fresenius Medical Care, reviewed the manuscript. However, the funding sources had no role in the preparation or analysis of data, and the decision to publish.

We note that you received funding from a commercial source: Fresenius Medical Care Deutschland GmbH

Please know it is PLOS ONE policy for corresponding authors to declare, on behalf of all authors, all potential competing interests for the purposes of transparency. PLOS defines a competing interest as anything that interferes with, or could reasonably be perceived as interfering with, the full and objective presentation, peer review, editorial decision-making, or publication of research or non-research articles submitted to oneof the journals. Competing interests can be financial or non-financial, professional, or personal. Competing interests can arise in relationship to an organization or another person. Please follow this link to our website for more details on competing interests: http://journals.plos.org/plosone/s/competing-interests

Please see below our updated Competing Interests Statement:

The studies included in this article were supported by an unrestricted grant from Fresenius Medical Care Deutschland GmbH (study NL33129.068.10 and NL43381.068.13), a Baxter International extramural grant (study NL35039.068.10; grant reference number 11CECHHDEU1004) and the Dutch Kidney Foundation (study NL35039.068.10; grant reference number SB166). BC and AG, who are employees of Fresenius Medical Care and hold shares in the company, reviewed the manuscript. However, the funding sources had no role in the preparation or analysis of data, and the decision to publish.

In addition, RJHM, NJHB and JPK are supported by an (un)restricted grant from Fresenius Medical Care. RJHM and NJHB received lectures fees from Fresenius Medical Care. MHC reports Astellas Travel support grants to visit the international conferences ESOT 2017 and TTS 2018.

This does not alter our adherence to PLOS ONE policies on sharing data and materials.

Data cannot be shared publicly because of privacy of research participants (i.e., data contain potentially identifiable patient information). Restrictions on sharing of such data are imposed by the EU General Data Protection Regulation (GDPR). Data are available for researchers who meet the criteria for access to confidential data, on reasonable request. Data can be requested via Maastricht UMC+, Dept. of Internal Medicine, Div. of Nephrology (secretariaat.nefrologie@mumc.nl) or the principal investigator, Prof. Dr. J. P. Kooman (jeroen.kooman@mumc.nl).

Not applicable.

Thank you for noticing. This part has been removed as part of the review process.

 

References

1. Doumatey AP, Zhou J, Adeyemo A, Rotimi C. High sensitivity C-reactive protein (Hs-CRP) remains highly stable in long-term archived human serum. Clin Biochem. 2014;47(4-5):315-8. Epub 2014/01/01. doi: 10.1016/j.clinbiochem.2013.12.014.

2. Schillinger M, Exner M, Mlekusch W, Haumer M, Ahmadi R, Rumpold H, et al. Balloon angioplasty and stent implantation induce a vascular inflammatory reaction. J Endovasc Ther. 2002;9(1):59-66. Epub 2002/04/18. doi: 10.1177/152660280200900111.

3. Jourde-Chiche N, Fakhouri F, Dou L, Bellien J, Burtey S, Frimat M, et al. Endothelium structure and function in kidney health and disease. Nature reviews Nephrology. 2019;15(2):87-108. Epub 2019/01/05. doi: 10.1038/s41581-018-0098-z.

4. Cobo G, Lindholm B, Stenvinkel P. Chronic inflammation in end-stage renal disease and dialysis. Nephrol Dial Transplant. 2018;33(suppl_3):iii35-iii40. Epub 2018/10/04. doi: 10.1093/ndt/gfy175.

5. Malyszko J. Mechanism of endothelial dysfunction in chronic kidney disease. Clin Chim Acta. 2010;411(19-20):1412-20. Epub 2010/07/06. doi: 10.1016/j.cca.2010.06.019.

6. Davies PF. Hemodynamic shear stress and the endothelium in cardiovascular pathophysiology. Nat Clin Pract Cardiovasc Med. 2009;6(1):16-26. Epub 2008/11/26. doi: 10.1038/ncpcardio1397.

7. Meeus F, Kourilsky O, Guerin AP, Gaudry C, Marchais SJ, London GM. Pathophysiology of cardiovascular disease in hemodialysis patients. Kidney Int Suppl. 2000;76:S140-7. Epub 2000/08/11.

8. Padilla J, Simmons GH, Bender SB, Arce-Esquivel AA, Whyte JJ, Laughlin MH. Vascular effects of exercise: endothelial adaptations beyond active muscle beds. Physiology (Bethesda). 2011;26(3):132-45. Epub 2011/06/15. doi: 10.1152/physiol.00052.2010.

---

## [Decision Letter · Decision Letter 1]

3 Sep 2019

[EXSCINDED]

Endothelial dysfunction and low-grade inflammation in the transition to renal replacement therapy

PONE-D-19-18004R1

Dear Dr. van Gennip,

We are pleased to inform you that your manuscript has been judged scientifically suitable for publication and will be formally accepted for publication once it complies with all outstanding technical requirements.

With kind regards,

Wisit Cheungpasitporn, MD, FACP, FASN

University of Mississippi Medical Center

Twitter: @wisit661 Email: wcheungpasitporn@gmail.com 

Academic Editor

PLOS ONE

Additional Editor Comments:

I want to commend the authors on their superb efforts to revise the manuscript according to all reviewers’ suggestions. The quality of the manuscript has improved substantially.

Reviewers' comments:

Reviewer's Responses to Questions

**Comments to the Author**

1. If the authors have adequately addressed your comments raised in a previous round of review and you feel that this manuscript is now acceptable for publication, you may indicate that here to bypass the “Comments to the Author” section, enter your conflict of interest statement in the “Confidential to Editor” section, and submit your "Accept" recommendation.

Reviewer #2: All comments have been addressed

Reviewer #3: All comments have been addressed

2. Is the manuscript technically sound, and do the data support the conclusions?

Reviewer #2: Yes

Reviewer #3: Yes

3. Has the statistical analysis been performed appropriately and rigorously? 

Reviewer #2: Yes

Reviewer #3: Yes

4. Have the authors made all data underlying the findings in their manuscript fully available?

Reviewer #2: Yes

Reviewer #3: Yes

5. Is the manuscript presented in an intelligible fashion and written in standard English?

Reviewer #2: Yes

Reviewer #3: Yes

6. Review Comments to the Author

Reviewer #2: (No Response)

Reviewer #3: Much improved manuscript from prior version. The paper, in its revised form, is suitable for publication

7. PLOS authors have the option to publish the peer review history of their article (what does this mean?). If published, this will include your full peer review and any attached files.

Reviewer #2: No

Reviewer #3: No

---

## [Editor Report · Acceptance letter]

6 Sep 2019

PONE-D-19-18004R1 

Endothelial dysfunction and low-grade inflammation in the transition to renal replacement therapy 

Dear Dr. Gennip:

I am pleased to inform you that your manuscript has been deemed suitable for publication in PLOS ONE. Congratulations! Your manuscript is now with our production department. 

With kind regards,

on behalf of

Dr. Wisit Cheungpasitporn 

Academic Editor

PLOS ONE